# EXAMPLE-BASED PLANNING VIA DUAL GRADIENT FIELDS

## ABSTRACT

Path planning is one of the key abilities of an intelligent agent. However, both the learning-based and sample-based planners remain to require explicitly defining the task by manually designing the reward function or optimisation objectives, which limits the scope of implementation. Formulating the path planning problem from a new perspective, *Example-based planning* is to find the most efficient path to increase the likelihood of the target distribution by giving a set of *target examples*. In this work, we introduce Dual Gradient Fields (DualGFs), an offline-learning example-based planning framework built upon score matching. There are two gradient fields in DualGFs: a *target gradient field* that guides task completion and a *support gradient field* that ensures moving with physical constraints. In the learning process, instead of interacting with the environment, the agents are trained with two offline examples, *i.e.*, the target gradients and support gradients are trained by *target examples* and *support examples*, respectively. The support examples are randomly sampled from free space, *i.e.*, states without collisions. DualGF is a weighted mixture of the two fields, combining the merits of the two fields together. To update the mixing ratio adaptively, we further propose a fields-balancing mechanism based on Lagrangian-Relaxation. Experimental results across four tasks (navigation, tracking, particle rearrangement, and room rearrangement) demonstrate the scalability and effectiveness of our method. Our codes and demonstrations can be found at https://sites.google.com/view/dualgf.

## 1 INTRODUCTION

Planning paths to reach a goal is a fundamental function of an intelligent agent (Russell, 2010) and has a wide range of real-world applications, such as navigation (Patle et al., 2019), object tracking (Zhong et al., 2019), and object rearrangement (King et al., 2016). Existing planning algorithms, whether sampling-based (LaValle et al., 1998a; Karaman & Frazzoli, 2011) or learning-based (Kulathunga, 2021; Yu et al., 2020; Tamar et al., 2016), need exhausted test-time sampling for searching a path or reward functions for learning. This severely limits the implementation scope of planning since many real-world tasks are hard to design the objectives/ reward with human priors, *e.g.*, tidying up a house, or rearranging a desktop.

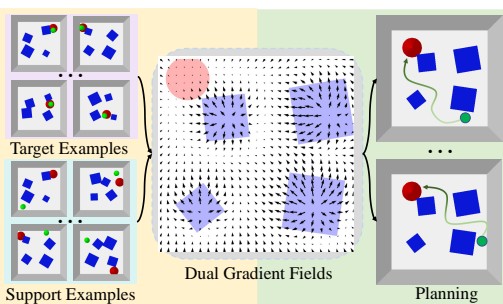

Figure 1: Our task setting. **Left:** The agent learns task specification from target examples and physical constraints from support examples during training. **Right:** The agent plans a path in novel conditions during the test phase.

In this paper, we consider a novel data-driven planning paradigm: *Example-based planning*, in which the user can specify the task by providing a set of *target examples*, rather than programming a task-specific objective. Benefiting from such a paradigm, example-based planning can scale to various tasks, particularly tasks with implicit goals, *i.e.* specifying the task with a target distribution instead of a specific goal state. Besides, the agent needs to infer the environmental constraints to safely move in a physical world. Previous approaches either learn physical constraints from interacting with the

environment and collision penalty (Wu et al., 2022), offline demonstrations (Janner et al., 2022), or exhaustively sampling points at test time (LaValle et al., 1998a). However, online interaction is costly and unsafe, while offline demonstrations are expensive to collect and test-time sampling is time inefficient.

To this end, we propose a fully example-based planning framework that learns two gradient fields with different purposes from examples by score-matching (Vincent, 2011), namely *DualGF*. DualGF consists of two fields: A *target gradient field* and a *support gradient field*. The target gradient field estimates the gradient of the target distribution so as to provide the fastest direction to accomplish the task. The support gradient field learns to reverse the perturbed state back to the free space so as to help avoid collisions. To combine the merits of the two fields, we further introduce a gradient mixer to adaptive balance the trade-off between the two gradients (keep safe vs. reach goal) when constructing the dual gradient field in execution. In practice, we can also incorporate the dual gradient field with a low-level controller to output primitive actions in control. The two gradients are trained from two sets of examples, respectively. For the target gradient field, we collect a set of target states sampled from the target distribution, such as a set of tidied rooms. For the support gradient field, we provide the agent with another set of examples (*support examples*) that are uniformly sampled from the free space, *i.e.*, states without collision. The support examples are abundant and easy to obtain in real scenarios, *e.g.*, randomly initialised objects, which largely alleviate the safety issue of learning from interactions. As illustrated in Fig. 1, the agent can learn generalisable inference of task specification and physical constraints from target and support examples and planning in *unseen* environment.

Our experiments validate the generalisation ability of the DualGF planning framework across a variety of tasks, including classical planning tasks such as navigation, tracking, and planning tasks without explicit goal specification such as object rearrangement (Wu et al., 2022). Specifically, the proposed DualGF significantly outperforms the learning-based baselines in planning performance and efficiency while achieving comparable performance with reference approaches that use the ground truth model or test-time sampling. Ablation studies also demonstrate the effectiveness of the proposed support gradient field and field-balancing mechanism.

In conclusion, our contributions are summarised as follows: a) We reformulate the path planning problem in a data-driven paradigm, where we specify the tasks with examples rather than manually design objectives; b) We propose a novel score-based planning framework DualGF that can adaptively integrate two gradient fields trained from different example sets so as to output instructions to complete a task; c) We conduct experiments in four tasks to demonstrate the scalability of our method, and empirical results show that DualGF significantly outperforms the state-of-the-art methods in efficiency and safety.

## 2 RELATED WORK

**Learning from Demonstration.** Example-based planning can be viewed as a special case of Learning from demonstration (LfD). LfD is a long-studied problem aiming at learning a policy from only a set of expert trajectories. There are two mainstreams of LfD: Behavioural Cloning (BC) (Pomerleau, 1991; Ross & Bagnell, 2010; Ross et al., 2011), which learns a policy in a supervised manner; and Inverse Reinforcement Learning (IRL) (Fu et al., 2017; Liu et al., 2020; Kostrikov et al., 2018; Ziebart et al., 2008), which finds a cost function under which the expert is uniquely optimal. Different from LfD, we train the agent from only two sets of examples instead of whole demonstrations. There are some RL-based methods that consider an example-based setting: VICE (Fu et al., 2018) is similar to AIRL (Fu et al., 2017), but is designed for learning from a set of success examples. SQIL (Reddy et al., 2019) is a method modified from SAC (Haarnoja et al., 2018) that labels success examples with a reward of +1. RCE (Eysenbach et al., 2021), a modification of the actor-critic-based method (Fujimoto et al., 2018; Haarnoja et al., 2018), directly learns a value function from transitions and successful examples. Recently, Offline-RCE (Hatch et al., 2022) extends RCE to an offline setting. Different from these methods, our method requires neither interaction with the environment nor offline demonstrations.

**Path Planning.** Sampling-based planning algorithms, such as the probabilistic roadmap method (PRM, (Kavraki & Latombe, 1994)) and the rapidly exploring random tree (RRT, (LaValle et al., 1998b)), are dominant in traditional path planning algorithms. PRT and its variants: (Dobson & Bekris, 2014); (You et al., 2021); (Chai et al., 2022); (Hüppi et al., 2022), intend to generate a

roadmap and find a collision-free path from a starting point to a goal region on the roadmap. RRT and its variants: (Strub & Gammell, 2020); (Gammell et al., 2020) (Li et al., 2022); (Strub & Gammell, 2022), iteratively build a tree by expanding towards the random sample point instead of directly connecting to the sample point. Different from these methods, our approach enables path planning in an end-to-end paradigm without test-time sampling, which is more time-efficient and scalable to high-dimensional tasks. Our method is also similar to artificial potential field Lee & Park (2003) which manually designs an attractive field guiding the agent to the goal location and a repulsive field pushing the agent away from the obstacles. Differently, our method learns two fields from two sets of examples that can generalise to unseen conditions. Besides, our method can plan with visual observation (See Appendix. D.4) and scale up to high-dimensional domains.

## 3 PROBLEM STATEMENT

The example-based planning can be described by a tuple $(\mathcal{S}, \mathcal{S}_f, p_{tar}, \rho, \epsilon)$ where $\mathcal{S}$ denotes the state space (*e.g.*, configurations of objects and obstacles) while $\mathcal{S}_f \subseteq \mathcal{S}$ denotes the free space (the valid states without object collision or transboundary). At each time step $t$, our planning agent $\pi : \mathcal{S} \to \mathcal{A}$ first outputs a target state within an epsilon-ball of the input state, *i.e.*, $||\hat{\mathbf{s}_{t+1}} - \mathbf{s}_t|| \leq \epsilon$. Then a simple low-level controller (*e.g.*, PID controller) outputs actions to achieve this state $\mathbf{a}_t = \mathcal{C}(\mathbf{s}_t, \hat{\mathbf{s}_{t+1}})$. During training, the agent is given a set of target examples $S^*_{tar} = \{\mathbf{s}^*, \mathbf{s}^* \sim p_{tar}(\mathbf{s})\}$ where $p_{tar} : \mathcal{S} \to \mathbb{R}^+$ denotes a target distribution. Similar to (Wu et al., 2022), the goal of the planning agent is to find the most efficient path to increase the likelihood of the target distribution:

$$\pi^* = \arg\max_{\pi} \mathbb{E}_{\rho(\mathbf{s}_0), \tau \sim \pi} \left[ \sum_{\mathbf{s}_t \in \tau} \gamma^t \log p_{tar}(\mathbf{s}_t) \right], \text{s.t.} \quad \forall \mathbf{s}_t \in \mathcal{S}_f \quad (1)$$

To this end, the agent needs to infer the task specification, *i.e.*, knowing where to move can probably succeed, as well as the free space, *i.e.*, avoiding collisions between objects and obstacles.

In this work, we consider the offline setting of example-based planning, where the agent has to infer physics constraints from support examples instead of interacting with the environment. The support examples are uniformly sampled from the free space $S^*_{sup} = \{\mathbf{s}^*, \mathbf{s}^* \sim p_{sup}(\mathbf{s})\}$ where

$$p_{sup}(\mathbf{s}) = \begin{cases} \dfrac{1}{|\mathcal{S}_f|}, & \mathbf{s} \in \mathcal{S}_f \\ 0, & \text{otherwise} \end{cases}, \quad |\mathcal{S}_f| = \int_{\mathcal{S}_f} ds \quad (2)$$

The rationale for 'replacing interactions with examples' is that the support examples are enough for inferring the free space $\mathcal{S}_f$ while the online interactions are unsafe and costly. Besides, the agent should learn generalisable inference of free space instead of overfitting to a specific environment.

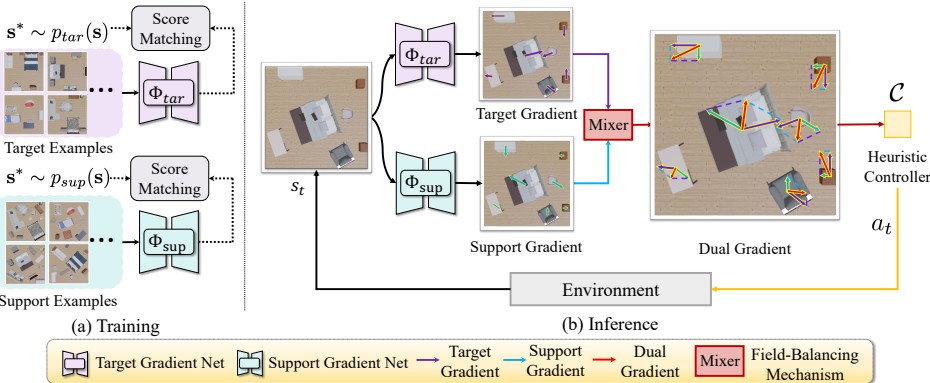

Figure 2: **Method Overview.** a): A target and a support score network are trained from examples via score-matching objective. b): DualGF computes target and support gradient at each time step, where the *target gradient* guides the task completion (*i.e.*, move objects to tidy layout) and the *support gradient* directs the objects to avoid collisions. DualGF adaptively mixes the two gradients via a field-balancing mechanism to output the dual gradient. Incorporated with a low-level controller, DualGF can interpret the dual gradient into an action.

## 4 METHODOLOGY

**Overview:** To tackle example-based planning, our key idea is to learn gradient fields from examples. As illustrated in Fig. 2 (a), the *target gradient field* $\Phi_{tar}^{\theta}$ and the *support gradient field* $\Phi_{sup}^{\phi}$ is trained from the target examples and support examples via score-matching respectively. The target gradient guides the task completion by providing the fastest direction to increase the task likelihood. The support gradient helps avoid collision by pointing to the inner of the free space. Finally, we construct a policy that plans the next state based on the mixture of the two gradients, *i.e.*, the *dual gradient*. In the control scenario, a heuristic-based controller can be used to interpret the dual gradient into a low-level action.

### 4.1 PLANNING WITH THE DUAL GRADIENT FIELDS

We tackle example-based planning by greedily maximising the target likelihood. Firstly, the agent aims at searching for the next state that maximises the target likelihood from the free space:

$$\underset{\mathbf{s}_{t+1}}{\operatorname{argmax}} \log p_{tar}(\mathbf{s}_{t+1}), \quad \text{s.t. } \log p_{sup}(\mathbf{s}_{t+1}) \geq \log \frac{1}{|\mathcal{S}_f|} \tag{3}$$

However, this objective is problematic since both the $\log p_{sup}(\mathbf{s}_{t+1})$ and $\log p_{tar}(\mathbf{s}_{t+1})$ are zero in many regions, such as non-target region or inner of the free space. Besides, $p_{sup}(\mathbf{s}_{t+1})$ is non-differentiable on the boundary of the free space $\partial \mathcal{S}_f$. As a result, we choose to 'soften' both distributions by adding a small Gaussian noise:

$$p_{sup}^{\sigma}(\hat{\mathbf{s}}) = \int \mathcal{N}(\hat{\mathbf{s}}; \mathbf{s}, \sigma^2 I) p_{sup}(\mathbf{s}) d\mathbf{s}, \quad p_{tar}^{\sigma}(\hat{\mathbf{s}}) = \int \mathcal{N}(\hat{\mathbf{s}}; \mathbf{s}, \sigma^2 I) p_{tar}(\mathbf{s}) d\mathbf{s} \tag{4}$$

In this way, we replace the objective in Eq. 3 with the following:

$$\underset{\mathbf{s}_{t+1}}{\operatorname{argmax}} \log p_{tar}^{\sigma}(\mathbf{s}_{t+1}), \quad \text{s.t. } \log p_{sup}^{\sigma}(\mathbf{s}_{t+1}) \geq c \tag{5}$$

where $c \in \mathbb{R}$ is a threshold indicating the conservative degree of the policy. The problem Eq. 5 can be further related to an unconstrained optimisation with a penalty term by the Lagrangian method:

$$\underset{\mathbf{s}_{t+1}}{\max} \ \underset{\lambda \geq 0}{\min} \log p_{tar}^{\sigma}(\mathbf{s}_{t+1}) + \lambda \left( \log p_{sup}^{\sigma}(\mathbf{s}_{t+1}) - c \right) \tag{6}$$

where $\lambda > 0$ is known as a Lagrangian multiplier. The $\mathbf{s}_{t+1}$ and $\lambda$ can be updated iteratively for multiple steps to obtain a solution. Here, we choose to update the $\mathbf{s}_{t+1}$ and $\lambda$ **only one step** for Eq. 6 since the $\hat{\mathbf{s}_{t+1}}$ is close to $\mathbf{s}_t$ in our problem, *i.e.*, $||\hat{\mathbf{s}_{t+1}} - \mathbf{s}_t||_2 \leq \epsilon$. :

$$\hat{\mathbf{s}_{t+1}} = \mathbf{s}_t + \mu_s \left( \nabla_{\mathbf{s}} \log p_{tar}^{\sigma}(\mathbf{s}_t) + \lambda_t \nabla_{\mathbf{s}} \log p_{sup}^{\sigma}(\mathbf{s}_t) \right)$$
$$\lambda_{t+1} = \text{ReLU} \left( \lambda_t - \mu_\lambda \left( \log p_{sup}^{\sigma}(\mathbf{s}_{t+1}) - c \right) \right) \tag{7}$$

where $\mu_s$ and $\mu_\lambda$ are two step sizes. Empirically, the latter is set as a fixed scalar $\mu_\lambda = 0.01$ while the former is properly set to ensure $||\mathbf{s}_{t+1} - \mathbf{s}_t||_2 \leq \epsilon$. In this way, our planning policy is derived as:

$$\hat{\mathbf{s}_{t+1}} = \mathbf{s}_t + \epsilon \frac{\mathbf{g}_{dual}}{||\mathbf{g}_{dual}||_2}, \quad \mathbf{g}_{dual} = \underbrace{\mathbf{g}_{tar} + \lambda_t \mathbf{g}_{sup}}_{dual \ gradient}$$
$$\mathbf{g}_{tar} = \underbrace{\nabla_{\mathbf{s}} \log p_{tar}^{\sigma}(\mathbf{s}_t)}_{target \ gradient}, \quad \mathbf{g}_{sup} = \underbrace{\nabla_{\mathbf{s}} \log p_{sup}^{\sigma}(\mathbf{s}_t)}_{support \ gradient} \tag{8}$$

Intuitively, the state is updated by the mixture of the *target gradient* $\nabla_{\mathbf{s}} \log p_{tar}^{\sigma}(\mathbf{s}_t)$ and the *support gradient* $\nabla_{\mathbf{s}} \log p_{sup}^{\sigma}(\mathbf{s}_t)$ weighted by $\lambda_t$. As shown in Fig. 3, the target gradient is a vector field that points to the goal region, so as to guide the task completion. The support gradient fields point to the inner region of the free space. The closer a state is to the boundary of the free space, the larger the magnitudes of its support gradient will 'push' it into the inner region. In Appendix. D.9, we also derive the closed-form of the target and support gradients and discuss the above insights.

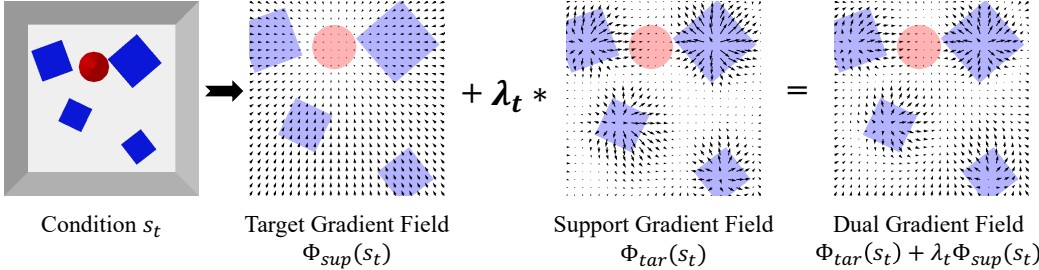

Condition $s_t$      Target Gradient Field $\Phi_{sup}(s_t)$      Support Gradient Field $\Phi_{tar}(s_t)$      Dual Gradient Field $\Phi_{tar}(s_t) + \lambda_t \Phi_{sup}(s_t)$

Figure 3: Visualisation of the learned target, support and dual gradient fields on *Navigation* task. The L2-norm of support gradients is small in the free space but grows significantly faster when close to the boundary. As a result, the direction of dual gradients is consistent with the target gradient in the free space and consistent with the support gradient near the boundary.

## 4.2 UPDATING MIXING RATE VIA SUPPORT GRADIENT FIELD

From Eq. 7, we can infer that $\lambda_t$ is updated to maintain the 'safety level' $\log p_{sup}^\sigma(\mathbf{s})$ at a certain threshold $c$, since a larger $\lambda_t$ will lead to an increase of $\log p_{sup}^\sigma(\mathbf{s})$ on the Further states, vice versa.

However, updating $\lambda_t$ requires the exact computation of $\log p_{sup}^\sigma(\mathbf{s}_{t+1}) - c$ which involves integration over the state space as shown in Eq. 4. This is intractable when the state space is high-dimensional. Thus, we seek to develop a practical field-balancing mechanism. We first approximate $\log p_{sup}^\sigma(\mathbf{s}_{t+1})$ from $\log p_{sup}^\sigma(\mathbf{s}_0)$ via first-order Taylor expansion:

$$
\begin{aligned}
\log p_{sup}^\sigma(\mathbf{s}_{t+1}) - c &= -\left(c - \log p_{sup}^\sigma(\mathbf{s}_0)\right) + \log p_{sup}^\sigma(\mathbf{s}_{t+1}) - \log p_{sup}^\sigma(\mathbf{s}_0) \\
&\approx -\left(c - \log p_{sup}^\sigma(\mathbf{s}_0)\right) + \sum_{k=0}^{t} \left\langle \nabla_{\mathbf{s}} \log p_{sup}^\sigma(\mathbf{s}_k), \mathbf{s}_{k+1} - \mathbf{s}_k \right\rangle
\end{aligned}
\tag{9}
$$

We further assume that $\mathbf{s}_0$ is always initialised far away from the boundary of the free space so that $\mathrm{Var}_{\mathbf{s}_0 \sim \rho(\mathbf{s})}[\log p_{sup}^\sigma(\mathbf{s})] \approx 0$. Thus, the remaining term $c - \log p_{sup}^\sigma(\mathbf{s}_0)$ can be regarded as a constant $c'$ in the sense of expectation:

$$
\mathbb{E}_{\mathbf{s}_0 \sim \rho(\mathbf{s})}\left[c - \log p_{sup}^\sigma(\mathbf{s}_0)\right] = c - \mathbb{E}_{\rho(\mathbf{s})}[\log p_{sup}^\sigma(\mathbf{s})] = c - \underbrace{H(\rho(\cdot), p_{sup}^\sigma(\cdot))}_{constant} \overset{def}{=} c'
\tag{10}
$$

## 4.3 INCORPORATING DUALGF WITH LOW-LEVEL CONTROLLER

Under the assumption that the system is holonomic, DualGF can also tackle the control tasks incorporated with a low-level controller $\mathcal{C} : \mathcal{S} \times \mathcal{S} \to \mathcal{A}$. For instance, if the atomic action space is velocity (with speed limit $v_{max}$ and simulation duration $\Delta t$), we can set $\epsilon = v_{max} \cdot \Delta t$ in Eq. 7 and cast the state change $(\mathbf{s}_{\hat{t+1}}, \mathbf{s}_t)$ into a target velocity via a heuristic-based controller:

$$
\mathbf{a}_t = \mathcal{C}(\mathbf{s}_{\hat{t+1}}, \mathbf{s}_t) = \frac{\mathbf{s}_{\hat{t+1}} - \mathbf{s}_t}{\Delta t} = \frac{v_{max}}{||\mathbf{g}_{dual}||_2} \cdot \mathbf{g}_{dual}
\tag{11}
$$

For the non-linear dynamics where the action space is forces imposed on the objects, we can leverage a PID controller to achieve the target velocity in multiple steps. In this work, we focus on the effectiveness of the high-level module trained under the fully example-based setting, *i.e.*, the dual gradient fields, instead of the complexity of the dynamics. Hence, we conduct experiments on tasks with velocity-based action space (*e.g.*, linear or angular velocity).

The whole planning framework is summarised below:

$$\hat{\mathbf{s}_{t+1}} = \mathbf{s}_t + \epsilon \frac{\mathbf{g}_{dual}}{||\mathbf{g}_{dual}||_2}, \quad \mathbf{g}_{dual} = \nabla_\mathbf{s} \log p^\sigma_{tar}(\mathbf{s}_t) + \lambda_t \nabla_\mathbf{s} \log p^\sigma_{sup}(\mathbf{s}_t)$$

$$\lambda_{t+1} = \text{ReLU} \left( \lambda_t - \mu_\lambda \left( \sum_{k=0}^t \left\langle \nabla_\mathbf{s} \log p^\sigma_{sup}(\mathbf{s}_k), \mathbf{s}_{k+1} - \mathbf{s}_k \right\rangle - c' \right) \right) \tag{12}$$

$$\mathbf{a}_t = \mathcal{C}(\hat{\mathbf{s}_{t+1}}, \mathbf{s}_t) = \frac{v_{max}}{||\mathbf{g}_{dual}||_2} \cdot \mathbf{g}_{dual}$$

### 4.4 LEARNING GRADIENT FIELDS FROM EXAMPLES

Finally, we seek to estimate the target and support gradient fields mentioned above to realise our planning framework. Thanks to score-based generative modelling, we can obtain a guaranteed estimation of these gradient fields from examples.

**Preliminary:** The Denoising Score-Matching (DSM) proposed by (Vincent, 2011) aims at estimating the *score function* of a data distribution $\nabla_\mathbf{s} \log p_{data}(\mathbf{s})$. Given a sample set $\{\mathbf{s}^* \sim p_{data}(\mathbf{s})\}$, DSM pre-specifies a noise distribution $q_\sigma(\widetilde{\mathbf{s}}|\mathbf{s})$, *e.g.*, $\mathcal{N}(0, \sigma^2 I)$, and trains a *score network* $\mathbf{\Phi}_\theta(\cdot)$ to denoise the perturbed data samples:

$$\mathcal{L}(\theta) = \mathbb{E}_{\widetilde{\mathbf{s}} \sim q_\sigma(\widetilde{\mathbf{s}}|\mathbf{s}), \mathbf{s} \sim p_{data}(\mathbf{s})} \left[ ||\mathbf{\Phi}_\theta(\widetilde{\mathbf{s}}) - \frac{\mathbf{s} - \widetilde{\mathbf{s}}}{\sigma^2}||_2^2 \right] \tag{13}$$

This objective guarantees the optimal score network satisfies $\mathbf{\Phi}_\theta^*(\mathbf{s}) = \nabla_\mathbf{s} q_\sigma(\mathbf{s})$ almost surely. When $\sigma$ is small enough, we have $\nabla_\mathbf{s} q_\sigma(\mathbf{s}) \approx \nabla_\mathbf{s} \log p_{data}(\mathbf{s})$, so that $\mathbf{\Phi}_\theta^*(\mathbf{s}) \approx \nabla_\mathbf{s} \log p_{data}(\mathbf{s})$.

**Training:** Adopting DSM, we train a *target score network* $\mathbf{\Phi}^\theta_{tar} : \mathbb{R}^{d_a+d_c} \to \mathbb{R}^{d_a}$ and a *support score network* $\mathbf{\Phi}^\phi_{sup} : \mathbb{R}^{d_a+d_c} \to \mathbb{R}^{d_a}$ from target examples $S^*_{tar}$ and support examples $S^*_{sup}$ respectively, where $d_a$ denotes the dimension of agent-state space (*e.g.*, agent's position and orientation) and $d_c$ denotes the dimension of conditional-state space (*e.g.*, agent's category, obstacles' state).

With $\mathbf{s}_a \in \mathbb{R}^{d_a}$ and $\mathbf{s}_c \in \mathbb{R}^{d_c}$ denoted as agent-state and conditional-state respectively ($\mathbf{s} = [\mathbf{s}_a, \mathbf{s}_c]$), the training objectives of both score networks are as follows:

$$\mathcal{L}(\theta) = \mathbb{E}_{q_\sigma(\widetilde{\mathbf{s}}_a|\mathbf{s}_a), p_{tar}(\mathbf{s})} \left[ ||\mathbf{\Phi}^\theta_{tar}([\widetilde{\mathbf{s}}_a, \mathbf{s}_c]) - \frac{\mathbf{s}_a - \widetilde{\mathbf{s}}_a}{\sigma^2}||_2^2 \right]$$

$$\mathcal{L}(\phi) = \mathbb{E}_{q_\sigma(\widetilde{\mathbf{s}}_a|\mathbf{s}_a), p_{sup}(\mathbf{s})} \left[ ||\mathbf{\Phi}^\phi_{sup}([\widetilde{\mathbf{s}}_a, \mathbf{s}_c]) - \frac{\mathbf{s}_a - \widetilde{\mathbf{s}}_a}{\sigma^2}||_2^2 \right] \tag{14}$$

We adopt a variant of DSM (Song et al., 2020) that conducts DSM under different noise scales simultaneously. We defer network architecture and key hyperparameters to Appendix. C.1.

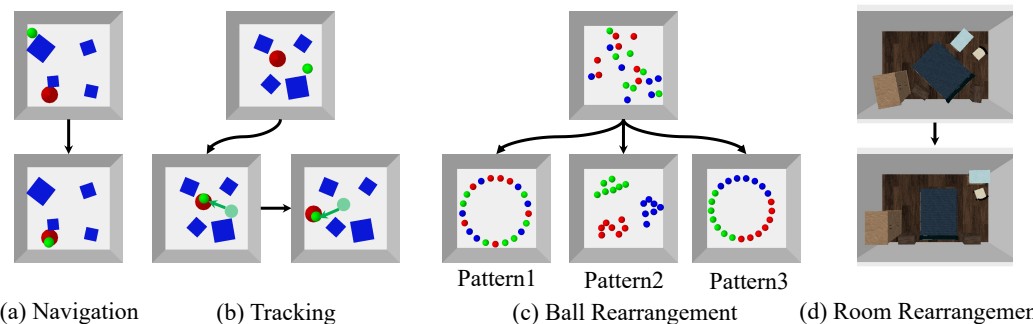

(a) Navigation      (b) Tracking      (c) Ball Rearrangement      (d) Room Rearrangement

Pattern1    Pattern2    Pattern3

Figure 4: Examples of four tasks used in the experiments. *Navigation* and *Tracking*: to reach/track a goal location while avoiding collision with obstacles. *Ball Rearrangement*: to rearrange balls into certain patterns. *Room rearrangement*: to rearrange the furniture into a reasonable layout.

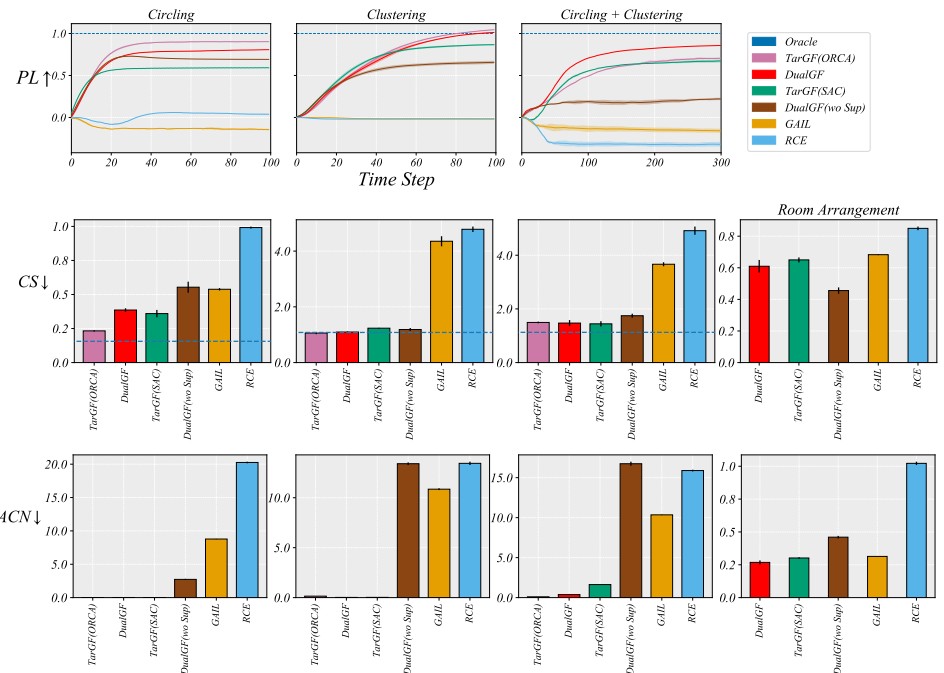

Figure 5: Comparative Results on *Ball Rearrangement* and *Room Rearrangement*. The results on *Ball Rearrangement* are plotted in the first three columns. The CS and ACN results on *Room Rearrangement* are plotted in the last column. The order of the X-axis is fixed across CS and ACN bars.

## 5 EXPERIMENT SETUPS

### 5.1 TASKS

We evaluate our method on three types of tasks with increasing difficulties. *Navigation* and *Tracking*: Classical path planning tasks with explicit task rewards. The agent needs to reach or tack a goal location while avoiding collision with the obstacles. The size and location of the obstacles changes in each episode. *Ball Rearrangement*: The agent needs to infer the pattern priors from different target examples and efficiently rearrange 21 balls into different types of patterns while covering all the modes of each pattern over the trajectories. *Room Rearrangement*: The agent learns arrangement priors from the target examples and rearranges the furniture in unseen layouts to cover the ground truth layout. The task demonstrations are illustrated in Fig. 4. We defer detailed state and action spaces, target and support examples, pseudo-likelihood function and more visualisations of the tasks to Appendix A.

### 5.2 EVALUATION

We collect trajectories over 5 random seeds for evaluation. For each task, we collect trajectories starting from the same initial states. We calculate the following metrics on the trajectories (for more details, we defer to Appendix. B): **Task Return weighted by Success (TRS)** calculates the averaged cumulative reward of an episode : $\mathbb{E}[\mathbb{1}(\tau \text{ is success}) \sum_{\mathbf{s}_t \in \tau} r_t]$ where $r_t$ denotes the immediate reward and $\mathbb{1}(\tau \text{ is success}) = 1$ only when $\tau$ has no collision. **Success Rate (SR)** calculates the percentage of non-collision trajectories. **Pseudo-Likelihood (PL)** measures the similarity between a given state and a target distribution by assigning a *pseudo-likelihood function* $\mathbf{F}_{proxy} : \mathcal{S} \to \mathbb{R}^+$. At each time step $t$, the PL-curve reports the averaged PL across all trajectories $\mathbb{E}[\mathbf{F}_{proxy}(\mathbf{s}_t)]$. **Coverage Score (CS):** reports the Minimal-Matching-Distance(MMD) (Achlioptas et al., 2018) between terminal states of a method $S_T$ and a fixed set of examples $S_{gt}$ from $p_{tar}$: $\sum_{\mathbf{s}_{gt} \in S_{gt}} \min_{\mathbf{s}_T \in S_T} ||\mathbf{s}_{gt} - \mathbf{s}_T||$.

**Averaged Collision Number (ACN)** reports the averaged collision number $\mathbb{E}[\mathbf{c}_t]$ at each time step where $\mathbf{c}_t = \sum c_{i,j}^t$, $c_{i,j}^t = 1$ only if the i-th and j-th object collide and $c_{i,j}^t = 0$ otherwise.

Table 1: Comparative results on *Navigation* and *Tracking*. The obstacles is always fixed in *Navigation (Static)* while changes for each episode in *Navigation (Dynamic)*.

| | | Navigation (Static) | | Navigation (Dynamic) | | Tracking | |
|---|---|---|---|---|---|---|---|
| | | TRS ↑ | SR ↑ | TRS ↑ | SR ↑ | TRS ↑ | SR ↑ |
| Learning Based | RL (SAC) | $9.1 \pm 0.6$ | $0.21 \pm 0.04$ | $8.3 \pm 0.4$ | $0.19 \pm 0.01$ | $0.0 \pm 0.0$ | $0.21 \pm 0.02$ |
| | GAIL | $1.5 \pm 0.4$ | $0.35 \pm 0.01$ | $1.6 \pm 1.0$ | $0.20 \pm 0.02$ | $0.1 \pm 0.1$ | $0.18 \pm 0.04$ |
| | RCE | $2.2 \pm 1.2$ | $0.13 \pm 0.01$ | $2.2 \pm 0.4$ | $0.22 \pm 0.04$ | $0.1 \pm 0.0$ | $0.17 \pm 0.03$ |
| | TarGF (SAC) | $7.0 \pm 0.2$ | $0.19 \pm 0.02$ | $4.4 \pm 0.3$ | $0.19 \pm 0.01$ | $0.2 \pm 0.1$ | $0.18 \pm 0.04$ |
| | DualGF (wo Sup) | $0.0 \pm 0.0$ | $0.07 \pm 0.01$ | $0.9 \pm 0.1$ | $0.09 \pm 0.03$ | $0.0 \pm 0.0$ | $0.11 \pm 0.04$ |
| | DualGF | $\mathbf{30.0 \pm 1.5}$ | $\mathbf{0.37 \pm 0.02}$ | $\mathbf{26.2 \pm 6.5}$ | $\mathbf{0.39 \pm 0.05}$ | $\mathbf{3.4 \pm 0.1}$ | $\mathbf{0.56 \pm 0.03}$ |
| Planning Based | PRM (Reference) | $37.8 \pm 3.6$ | $0.45 \pm 0.04$ | $39.5 \pm 3.3$ | $0.47 \pm 0.04$ | $4.1 \pm 0.9$ | $0.16 \pm 0.04$ |
| | Potential Field | $34.8 \pm 0.8$ | $0.45 \pm 0.00$ | $23.2 \pm 5.6$ | $0.28 \pm 0.07$ | $2.2 \pm 0.6$ | $0.09 \pm 0.02$ |

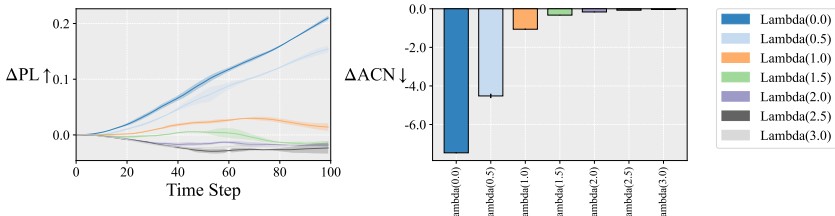

Figure 6: Ablation results on *Clustering*, where $\Delta$Metric $=$ Metric(DualGF) - Metric(Ablated). With mixing rate $\lambda_0$ decreasing, the advantage of *DualGF* over the ablation increases significantly.

The primary metrics for each task: TRS and SR for *Navigation* and *Tracking*; PL and ACN for *Ball Rearrangement*; CS and ACN for *Room Rearrangement*.

## 5.3 BASELINES

We compare our framework (*i.e.*, *DualGF*) with the following learning-based baselines and planning-based baselines. The implementation details are deferred to Appendix. C.2. **Leaning-based Baselines:** *TarGF(SAC):* is the learning-based framework of TarGF (Wu et al., 2022). *GAIL:* A classical inverse RL method that trains a discriminator as the reward function. This method trains two classifiers from target and support examples, respectively. *RCE:* The SOTA example-based reinforcement learning method. *RL(SAC):* The agent is trained under the ground-truth task reward and collision signals. **Planning-based Baselines:** *TarGF(ORCA):* A model-based framework that requires the ground-truth model. *Probabilistic Road Map (PRM):* A classical sampling-based method Kavraki & Latombe (1994). *Potentential Field:* Artificial potential field method Lee & Park (2003). **Oracle:** The oracle performance is obtained by slightly perturbing the target examples with a small Gaussian noise.

## 6 EXPERIMENT RESULTS

### 6.1 BASELINE COMPARISON

As seen from the quantitative result in Tab. 1 and Fig. 5, *DualGF* significantly outperforms baselines and achieves comparable performance with reference methods such as *TarGF(ORCA)* and *PRM*:

**In *Navigation and Tracking*:** 1) *DualGF* outperforms learning-based baselines by a large margin. The classical planning algorithm *PRM* achieves strong results in task reward (TR) since it can search for a near-optimal path without any collision to reach the goal in most cases. However, even if we have provided an exhaustive sampling budget (*i.e.*, 2000) for *PRM* in each episode, *PRM* still fails to search for a solution in some cases. As a result, *DualGF* outperforms *PRM* in ACN on *Navigation (Static)* and *Tracking* due to the failure cases of *PRM*. 2) Notably, the *PRM* is of poor time efficiency compared with *DualGF*. As shown in Tab. 2, *PRM* takes about 6 times of *DualGF* in *Navigation* and 110 times in *Tracking* since *PRM* needs to search a path again when the agent reaches the goal. 3) The *Navigation and Tracking* is non-trivial. The vanilla reinforcement learning approach *RL(SAC)* ranks second in the simplest task *Navigation (Static)*, yet it fails to keep the advantage in more challenging tasks, *i.e.*, *Navigation (Dynamic)* and *Tracking*.

**In *Ball Rearrangement*:** 1) As Fig. 5 illustrated, *DualGF* achieves the best performance compared with learning-based baselines in PL and ACN. In the hardest task *Circling + Clustering*, *DualGF* even outperforms than *TarGF(ORCA)* that plans with the ground truth model. These demonstrate the effectiveness of our method to scale to high-dimensional tasks. 2) The CS of *DualGF* is comparable with other competitive baselines such as *TarGF(SAC)*. Hence, our method is not trivially overfitting to a single mode.
3) The classifier-based methods (*i.e.*, *GAIL* and *RCE*) fail due to the training collapse of the classifier. In practice, we observe that classifier-based baselines achieve appealing results during the training process yet fail to converge at a high-performance level due to the over-exploitation of the classifier.

**In *Room Rearrangement*:** 1) *DualGF* achieves the best performance compared with baselines in both CS and ACN. This indicates that our method can generalise well

|        | Milliseconds per Step | |
|--------|------------|----------|
|        | Navigation | Tracking |
| DualGF | **3.3 ± 0.2** | **3.4 ± 0.2** |
| PRM    | 19.7 ± 2.3 | 376 ± 35.9 |

Table 2: The inference time per step averaged over 100 trajectories.

to the diverse conditional attributes such as different sizes, numbers and composition of objects. 2) The planning-based methods either suffer from scalability issues that *PRM* and Rapidly-exploring Random Trees (RRT) (LaValle et al., 1998a) cannot search for a feasible solution in 10 minutes. Besides, *TarGF(ORCA)* assumes the agent is circular in shape which is infeasible in *Room Rearrangement*. Hence, we do not compare them in *Room Rearrangement*.

## 6.2 ABLATION STUDIES AND ANALYSIS

We further conduct ablation studies to analyse the effectiveness of the key components of our method, *i.e.*, the *support gradient field* and the *field-balancing mechanism*. Further ablations on key hyperparameters (*i.e.*, initial $\lambda$ and $c'$), size of the target and support sets and the choice of the noise level $t$ are deferred to Appendix D.1, D.2 and D.3 respectively.

**Effectiveness of the Support Gradient Field:** *Ours wo Sup* is the ablated version that fixes the mixing rate to zero $\lambda \equiv 0$. 1) The task performance of *DualGF wo Sup* is worse than *DualGF* in all tasks across both *Navigation and Tracking* and *Ball Rearrangement*. Besides, the ACN of *DualGF* is significantly lower than *DualGF wo Sup* These results indicate that the support gradient field can help avoid collision effectively and further improve task performance. 2) In *Room Rearrangement*, *DualGF wo Sup* achieves better CS than *DualGF*, yet performs worse than baselines in ACN. Meanwhile, *DualGF* outperforms the baselines in both CS and ACN. This indicates support gradient field helps to find a better trade-off between task completion and collision avoidance.

**Effectiveness of Field-balancing Mechanism:** *DualGF(Fixed)* is the ablated version of *DualGF* with mixing rate lambda is fixed at an initial level. We illustrate the advantage of *DualGF* over *DualGF(Fixed)* on PL and ACN on Fig. 6. With the initial lambda decrease, the advantage of *DualGF* increases significantly in both PL and ACN. This indicates our field balancing mechanism can moderate the sensitivity to the choice of initial mixing rate $\lambda_0$.

## 7 CONCLUSION

In this study, we first reformulate the path planning problem in a novel data-driven paradigm where the agent learns to plan with two sets of examples. We further propose a novel score-based planning framework (DualGF) to tackle this problem. There are two gradient fields in DualGFs: a *target gradient field* that guides task completion and a *support gradient field* that ensures moving with physical constraints. Moreover, an adaptive gradient balance mechanism is introduced to combine the merits of the two fields. Experiments demonstrate the scalability of our method, and empirical results show that it significantly outperforms state-of-the-art baselines in efficiency and safety.

**Limitations and Future Work:** The main limitations of this work are the local minimum issue as discussed in Appendix D.6 and the holonomic assumption made in Sec 4.3. Another minor issue is the inherent instability of the Lagrangian Relaxation which can be mitigated via a small learning rate and gradient clipping, as discussed in Appendix D.7. In the future, we may incorporate our method with MCMC-based approaches such as Langevin Dynamics to overcome the local minimum issues. Besides, we may extend our method to non-holonomic scenarios by integrating model-predictive control (MPC).

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

# A TASK DETAILS

## A.1 TRACKING AND NAVIGATION

**Navigation and Tracking** requires the agent to reach a static location or track a dynamic location while avoiding collision with the obstacles. There are four rectangular obstacles and a particle agent. The scale, position and orientation of the obstacles are randomised before each episode in both tracking and navigation. The goal location is randomly reset at the start of the episode. Note that in tracking, the goal location will be further reset when reached by the agent. In this task, the target distribution is assigned as a Gaussian distribution centred on the goal location, *i.e.*, $\mathcal{N}(\mathbf{g}, 0.1^2 I)$. The state space is the concatenation of the agent's position, goal location and geometry observations of obstacles. The action space is two-dimensional linear velocity.

**State and Action Spaces:** $\mathbf{s} = [\mathbf{s}_a, \mathbf{s}_c]$ where the agent state is 2-D location $\mathbf{s}_a = [x, y] \in [-1, 1]^2$ and condition-state is the 2D goal location and obstacles' states $\mathbf{s}_c = [g, o_1, o_2, o_3, o_4]$ where $g \in [-1, 1]^2$ and $o_i$ denotes the state of the i-th obstacle. The obstacle state is the normalised position, orientation and side-length $o_i = [pos, ori, side]$ where $pos \in [-1, 1]^2, ori = [sin(yaw), cos(yaw)] \in [-1, 1]^2, side \in [-1, 1]$.

**Horizon:** 100.

**Initial Distribution:** Before each episode, we first randomly initialise 4 box-shaped obstacles and then randomly sample the agent's state that does not collide with the obstacles.

**Dynamics:** The floor and wall are all absolutely smooth planes. The arena is a 0.25m x 0.25m area, with the agent's radius being 0.015m. We set the friction coefficients of all obstacles and agents to 10.0 to penalise the collision.

**Target Examples:** To sample a target example, we first randomly initialise the obstacles' locations and sizes and the goal location. Then, we sample an agent's location via repeatedly sampling from a Gaussian distribution centred on the fixed goal location until the location is collision-free. To ensure the target score network can work well, we sample 100,000 examples for training.

**Support Examples:** To sample a support example, we first randomly initialise the obstacles' locations and sizes and the goal location. Then, we sample an agent's location via repeatedly sampling from the uniform distribution over the configuration space until the location is collision-free. To collect support examples, we reset the environment 200 times. For each reset, we collect 1000 examples from the same condition. Thus, the number of support examples is 200,000 in total.

## A.2 BALL REARRANGEMENT

**Ball Rearrangement** includes three sub-tasks: *Circling*, *Clustering*, and the hybrid of the first two, *Circling + Clustering*. In both sub-tasks, there are three sets of 7 balls that belong to different categories, *i.e.*, red, green and blue. The state and action spaces are the joint positions and linear velocities of all balls respectively. The agent is required to rearrange all balls to maximise the target likelihood of the joint state while avoiding collisions. The interpretations of the target distribution of each task are as follows: *Clustering* requires all balls to form into three clusters by colour; Both *Circling* and *Circling + Clustering* require all balls to form into a circle that can be centred anywhere in the arena, except that the latter further requires that the balls of the same type are adjacent to each other.

**State and Action Spaces:** There are 21 balls in the environment. The state space is the concatenation of all sub-states of all balls: $\mathbf{s} = [\mathbf{s}_1, \mathbf{s}_2, ..., \mathbf{s}_2 1]$. For each ball, the sub-state is $\mathbf{s}_i = [\mathbf{s}_a, \mathbf{s}_c], \mathbf{s}_a = [x, y], \mathbf{s}_c = [c]$ where $x, y \in [-1, 1]$ denotes the two dimensional position and $c \in \{0, 1, 2\}$ is a class label. The action space is also the concatenation of all sub-actions of all balls: $\mathbf{a} = [\mathbf{a}_1, \mathbf{a}_2, ..., \mathbf{a}_2 1]$ where $\mathbf{a}_i \in [-1, 1]^2$ is a 2-dimensional linear velocity.

**Horizon:** 300 for *Circling + Clustering* and 100 for *Circling* and *Clustering*.

**Initial Distribution:** We first uniformly sample rough locations for each ball, and then we eliminate overlaps between these positions by executing the physical simulation.

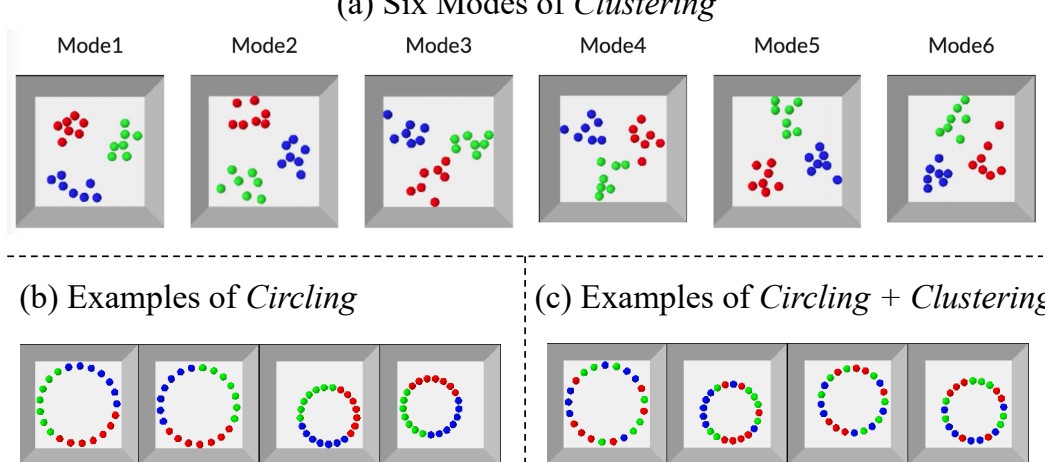

Figure 7: Visualisation of three sub-tasks of *Ball Rearrangement*. **(a):** *Clustering* requires all balls to form into three clusters by colour. **(b):** *Circling* requires all balls to form into a circle, neglecting the colour order. **(c):** *Circling + Clustering* require all balls to form into a circle while the balls of the same type are adjacent to each other.

**Dynamics:** The floor and wall are all absolutely smooth planes. All the balls are bounded in an 0.3m x 0.3m area, with the radius being 0.025m. We set the friction coefficients of all balls to 100.0 since we observe that setting a small(*e.g.*not larger than 1.0) friction coefficient does not significantly affect the dynamics. Besides, to increase the complexity of the dynamics, we set the masses and restitution coefficients of all green and blue balls to 0.1 and 0.99, respectively. The masses and restitution coefficients of all red balls are set to 10 and 0.1, respectively. We observe that under these dynamics, the collision may significantly harm the efficiency of the rearrangement process. Hence, the agent has to adapt to the dynamics for more efficient object rearrangement.

**Target Examples:** For *Circling*, first random sample a feasible centre in the free space. Given the centre, we further random sample a feasible radius for the target circle. Then we sample a circle orientation of the target circle. Given the centre, radius and orientation, we can obtain feasible locations for balls to form a circle. Finally, we randomly assign these positions to the balls. For *Clustering*, we directly sample initial ball locations from a Gaussian Mixture Model in Eq. 15. Then we step physical simulations to remove the collision between balls. The final state serves as a target example. For *Circling+Clustering*, it is similar to *Circling*. We first sample feasible locations for balls to form in a circle. Then we randomly sample a 'colourisation starting point'. Starting from this point, we colourise the balls in $R - G - B$ or $R - B - G$ order (50% for each order). For these three tasks, we collect 100,000 target examples. However, we demonstrate in Appendix D.2 that the target examples can be reduced to 100 in *Clustering*.

**Support Examples:** When sampling support examples, we increase the number of balls from $3 \times 7$ to $3 \times 10$ so as to increase the ball density in the arena. Since our score network is implemented as a graph neural network (As mentioned in Appendix C.1), the network can be transferred to different dimensions of input. To sample a support example, we uniformly sample locations for all the balls independently. Then we step physical simulations to remove collisions between balls to obtain a support example. To collect support examples, we repeat the above procedure to obtain 100, 000 examples. As shown in Appendix D.2, 50,000 support examples also work well.

### A.3 ROOM REARRANGEMENT

**Room Rearrangement** is built on a large-scale, synthetic indoor scene dataset 3D-FRONT (Fu et al., 2021). Following (Wu et al., 2022), we use 756 of 839 rooms for training and 83 for testing. The agent is required to learn scene priors from the target examples (*i.e.*, training set) and rearrange

objects into reasonable layouts during the test phase. The state space is the concatenation of observations of each object and the room boundary information. The action space is the concatenation of the two-dimensional linear velocity and one-dimensional angular velocity of each object.

**Dataset and Simulator:** We clean the 3D-Front dataset Fu et al. (2021) to obtain bedrooms that consist of four walls and three to eight objects. We augment each room by flipping two times and rotating four times to get eight augmentations. We then import these rooms into iGibson Shen et al. (2021) to run the physical simulation. The 756x8 rooms are used for *target examples* that are used for training the target score network, the classifier-based baselines and the VAE in goal-conditioned baselines. The other 83x8 rooms are used to initialise the room in the test phase, so the room rearrangement task is performed on the test dataset. The rooms in the training dataset are only used for learning prior knowledge to arrange the room.

**State and Action Spaces:** The state consists of a aspect ratio $r_a \in \mathbb{R}^+$ and an object state $\mathbf{s}_o \in \mathbb{R}^{K \times 6}$ where $K$ denotes the number of objects. The aspect ratio $r_a = \tanh(\frac{b_x}{b_y})$ where $b_x$ and $b_y$ denotes the horizontal and vertical wall bounds. The object state is the concatenation of sub-states of all the objects $\mathbf{s}_o = [\mathbf{s}^1, \mathbf{s}^2, ...\mathbf{s}^i, ..., \mathbf{s}^K]$ where the sub-state of the i-th object $\mathbf{s}^i \in \mathbb{R}^6$ is consists of 2-D position, 1-D orientation, 2-D bounding bound and a 1-D category label. The action is also a concatenation of sub-actions of all the objects $\mathbf{a}_o = [\mathbf{a}^1, \mathbf{a}^2, ...\mathbf{a}^i, ..., \mathbf{a}^K]$. For the i-th object, the action $\mathbf{a}^i \in \mathbb{R}^3$ consists of a 2-D linear and a 1-D angular velocity. The whole action space is normalised into a $3 \times K$ dimensional unit-box $[-1, 1]^{3 \times K}$ by the velocity bounds. Note that the agent-state $\mathbf{s}_a$ in this task is the concatenation of agents' positions and orientations while the condition-state $\mathbf{s}_c$ in this task is the aspect ratio and the concatenation of agents' labels and 2-d bounding boxes.

**Horizon:** Each training episode contains 100 steps, instead of 250 steps used in Wu et al. (2022).

**Initial Distribution:** To guarantee the initial state is accessible to the high-density region of the target distribution, we sample an initial state in two stages: First, we sample a room from the 83x8 rooms in the test dataset. Then we perturb this room by 1000 Brownian steps.

**Dynamics:** For more efficient environment reset and physical simulation, we build a 'proxy simulator' based on PyBullet Coumans & Bai (2016–2021) to replace the iGibson simulator. We use iGibson to load and save the metadata of each room. Then we reload these rooms in the proxy simulator, where each object is replaced by a box with the same geometry. We set the friction coefficient of all the objects in the room to zero, as the dynamics of the room are complex enough.

**Target Examples:** We use the 756x8 rooms from the training set as target examples.

**Support Examples:** Starting from each state in the target examples, we perturb the state via Brownian steps. For every 800 steps, we collect a state and then step simulations to turn it into a feasible support example. For each room, we collect 100 times. After cleaning the data, we obtain a support set with $448, 754$ examples.

## B    METRIC DETAILS

Here we introduce additional details of PL and CS.

### B.1    PSEUDO-LIKELIHOOD (PL)

measures the similarity between a given state and a target distribution by assigning a *pseudo-likelihood function* $\mathbf{F}_{proxy} : \mathcal{S} \rightarrow \mathbb{R}^+$. At each time step $t$, the PL-curve reports the averaged PL across all trajectories $\mathbb{E}[\mathbf{F}_{proxy}(\mathbf{s}_t)]$. Hence, the PL curve indicates the efficiency and final performance of each method. We do not report the PL-curves on *Room Rearrangement*, since it is hard to describe the human preferences and scene priors by hand-engineering.

The PL curve is only reported in *Ball Rearrangement*. The specific pseudo-likelihood functions are listed as follows:

**Circling:** The pseudo-likelihood function is defined as $\mathbf{F}_{proxy}(\mathbf{s}) = \exp^{-(\sigma_\theta + \sigma_r)}$, where $\sigma_\theta$ and $\sigma_r$ denote the standard deviation of the angle between two adjacent balls and the distances from each

ball to the centre of gravity of all balls, respectively. Intuitively, if a set of balls are arranged into a circle, then the $\sigma_r$ and $\sigma_\theta$ should be close to zero, achieving higher pseudo-likelihood.

**Clustering:** Different from (Wu et al., 2022), our target distribution is a six-mode Gaussian instead of a two-mode Gaussian:

Defining the joint centres' positions as a latent variable $C = (C_r, C_g, C_b)$ where $C_r$, $C_g$ and $C_b$ denote centres of red, green and blue balls respectively and the above six modes as $\{c_i\}_{1 \leq i \leq 6}$, the $C$ obeys a categorical distribution $p(C = c_i) = \frac{1}{6}$. The pseudo-likelihood function is a Gaussian Mixture Model:

$$p_{GMM}(\mathbf{s}) = \sum_{1 \leq k \leq 6} p(C = c_k) p(\mathbf{s}|C = c_k)$$

$$p(\mathbf{s}|C = c_k) = \prod_{1 \leq i \leq \frac{K}{3}} \mathcal{N}(C_r^k, 0.05I)(\mathbf{s}^i) \prod_{\frac{K}{3} \leq i \leq \frac{2K}{3}} \mathcal{N}(C_g^k, 0.05I)(\mathbf{s}^i) \prod_{\frac{2K}{3} \leq i \leq K} \mathcal{N}(C_b^k, 0.05I)(\mathbf{s}^i)$$

(15)

**Circling+Clusterng:** The pseudo-likelihood function is defined as $\mathbf{F}_{proxy}(\mathbf{s}) = \exp^{-(\sigma_\theta + \sigma_r)} \cdot \exp^{-(\sigma_R + \sigma_G + \sigma_B) - \sigma_C}$ where $\sigma_R$ denotes the standard deviation of the angle between two adjacent red balls, and $\sigma_G$ and $\sigma_B$ and $\sigma_C$ denotes the standard deviation of the positions of red, green and blue centres. Intuitively, the first term $\exp^{-(\sigma_\theta + \sigma_r)}$ measures the pseudo-likelihood of balls forming a circle and the next term $\exp^{-(\sigma_R + \sigma_G + \sigma_B) + \sigma_C}$ measures the pseudo-likelihood of balls being clustered into three piles.

### B.2 COVERAGE SCORE (CS):

CS measures the diversity and fidelity of the terminal states $S_T =$. CS reports the Minimal-Matching-Distance(MMD) (Achlioptas et al., 2018) between $S_T$ and a fixed set of examples $S_{gt}$ from $p_{tar}$: $\sum_{\mathbf{s}_{gt} \in S_{gt}} \min_{\mathbf{s}_T \in S_T} ||\mathbf{s}_{gt} - \mathbf{s}_T||$. This metric can detect mode-collapsing: If an agent only moves objects to a single mode, the terminal states may be far away from other modes which leads to a high value of CS (*i.e.*, bad performance).

**For Ball Rearrangement:** To calculate the coverage score, we sample fixed sets examples from the target distribution serving as $S_{gt}$ for *Circling*, *Clustering*, and *Circling+Clustering*, respectively. We sample 20 examples for *Circling* and *Circling+Clustering* and 50 examples for *Clustering*. Since the balls in the same category can actually be viewed as a two-dimensional point cloud, we measure the distance between two states by summing the CDs between each pile of balls by category.

**For Room Rearrangement:** The coverage score is calculated by averaging the coverage score in each room condition since the state dimension differs in different rooms. For each room in 83 test rooms, we calculate the coverage score between the eight ground truth states and eight rearrangement results and then the averaged coverage score over the 83 rooms is taken as the final coverage score for a method. We measure the distance between two states by calculating the average L2 distance between the positions of the corresponding objects.

## C IMPLEMENTATION DETAILS

For all learning-based baselines, we implement all the methods including ours in the same network architecture design and basically the same capacity (*e.g.*, hidden dimensions).

### C.1 DUALGF

*Training Objective:* The complete training objective for both target and support networks is the SDE-based score-matching objective proposed by Song et al. (2020):

$$\mathbb{E}_{t \sim \mathcal{U}(0,1)} \mathbb{E}_{\mathbf{s}(0) \sim p_{data}(\mathbf{s})} \mathbb{E}_{\mathbf{s}(t) \sim p_{0t}(\mathbf{s}(t)|\mathbf{s}(0))} [\mathbf{\Phi}_\theta(\mathbf{s}(t), t) - \nabla_{\mathbf{s}(t)} \log p_{0t}(\mathbf{s}(t) \mid \mathbf{s}(0))||_2^2].$$

(16)

where $p_{0t}(\mathbf{s}(t) \mid \mathbf{s}(0)) = \mathcal{N}(\mathbf{s}(t); \mathbf{s}(0), \frac{1}{2 \log \sigma}(\sigma^{2t} - 1)\mathbf{I})$ and $\sigma = 25$ is a hyper-parameter.

Using this objective, we can obtain the estimated score *w.r.t.* different levels of the noise-perturbed target distribution $p_{data}^t(\mathbf{s}(t)) = \int p_{0t}(\mathbf{s}(t) \mid \mathbf{s}(0))p_{data}(\mathbf{s}(0))d\mathbf{s}(0)$ simultaneously. In this way, we can assign different noise-level $t$ for different tasks conveniently. For all tasks, our method chooses $t = 0.01$.

**Network Architecture:** In *Navigation* and *Tracking*, our score networks are simply implemented as fully connected layers. We first encode the agent position $\mathbf{s}_a$ and noise-level $t$ into feature vectors $f_a$ and $f_t$. The state of each obstacle $o_i$ is encoded by a shared linear layer to $f_{o_i}$. We then pool the $\{f_{o_i}\}_i^4$ into a single vector. The support score network concatenates the $f_a$, $f_t$ and $f_o$ and sends them to a linear fusion layer to get the score output. The target score network further encodes goal location into $f_g$ and concatenates the $f_a$, $f_t$, $f_g$ and $f_o$ and sends them to a linear fusion layer to get the score output. In *Ball Rearrangement* and *Room Rearrangement*, the architecture of the target score network is exactly the same as the score network of Wu et al. (2022) used in *Clustering* and *Cirlcing+Clustering*. The support score network is exactly the same as the score network of Wu et al. (2022) used in *Circling* which does not encode the class label into the initial feature.

**Reproduction:** To entirely reproduce the results and check the details, we recommend running our code released on the Anonymous Github `https://anonymous.4open.science/r/ICLR23AnonymousCode-1426`.

## C.2 BASELINES

## C.3 TARGF

The training objective of TarGF is the same as Eq. 16. And we set $t = 0.01$ for all experiments for a fair comparison.

**For TarGF(SAC):** The implementation is based on the codes released by authors, except that we set $t = 0.01$ for both reward estimation and residual policy for a fair comparison.

**For TarGF(ORCA):** We set $\tau = 0.1$ and the simulation duration of each timestep $\Delta t = 0.02$. For each agent(object), ORCA only considers the 2-nearest agents as neighbours, since in our experiments ORCA often has no solution when the number of neighbours is larger than 2.

## C.4 CLASSIFIER-BASED BASELINES

These baselines refer to the *RCE* and *GAIL* in experiments.

*RCE* and is implemented based on the codes Eysenbach et al. (2021) released by RCE's authors. We only modify $\gamma = 0.95$, the training steps decrease to 0.5 million (*i.e.*, the same number of training steps as other methods) and the model architecture. The architecture of actor and critic networks is implemented the same as ours(*i.e.*, the same feature extraction layers and edge-convolutional layers).

*GAIL* trains two classifiers by discriminating two example sets with agent's rollouts respectively. The discriminator takes only one state instead of two adjacent states as input since the 'ground truth' reward(*i.e.*, likelihood) is defined on the current state. At each training step, we update the discriminator by distinguishing between the agents' and the expert's states(for one step) and then update the RL policy under the reward given by the discriminator(for one step).

## C.5 RL(SAC) IN NAVIGATION AND TRACKING

This is implemented as ablation of *GAIL* that replaces the classifiers with ground truth task rewards.

## C.6 PRM

We implement PRM from an off-the-shelf open-sourced repository PythonRobotics Sakai et al. (2018) with 16.3k stars. This implementation is also used for navigation tasks. We keep the default configuration and properly rescale our configurations to fit the API.

### C.7 POTENTIAL FIELD

This baseline is also implemented from an off-the-shelf open-sourced repository Python-Robotics Sakai et al. (2018) with 16.3k stars. We set $\zeta = 1.0$ for the attractive field and $\zeta = 1.0$ for the repulsive field. The distance threshold of the repulsive field is set to be $Q^* = 0.1$.

## D  ADDITIONAL ANALYSIS

### D.1  ABLATIONS ON MIXING RATE $\lambda$ AND THRESHOLD $c'$

We seek to evaluate the sensitivity of our method to the two key hyperparameters, *i.e.*, $\lambda_0$ and $c'$. The experiments are conducted on *Clustering* for better quantitative evaluation.

We evaluate *DualGF* with different levels of the initial lambda $\lambda_0$. As shown in Fig. 8, the PL and CS do not significantly change when the $\lambda \geq 1.0$. This indicates our method is not sensitive to $\lambda$ when it is above a positive threshold.

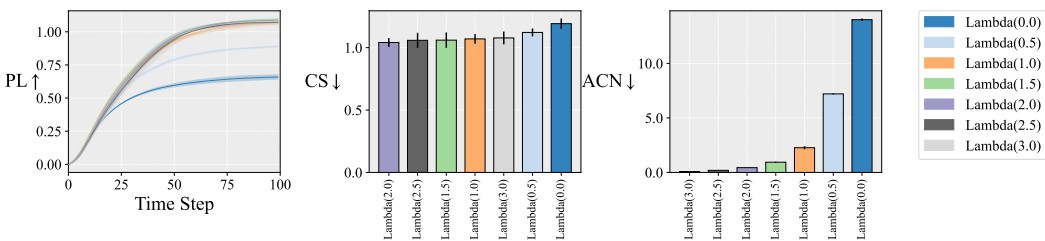

Figure 8: Ablation study results on the mixing rate $\lambda_t$. We fix the threshold $c' = 8$ and evaluate DualGF with different initial lambda.

Further, we evaluate *DualGF* with different levels of the initial threshold $c'$. As shown in Fig. 9, the PL and CS do not significantly change with all different levels of threshold but the ACN becomes better with the increase of the threshold. This indicates our method is not sensitive to $c'$.

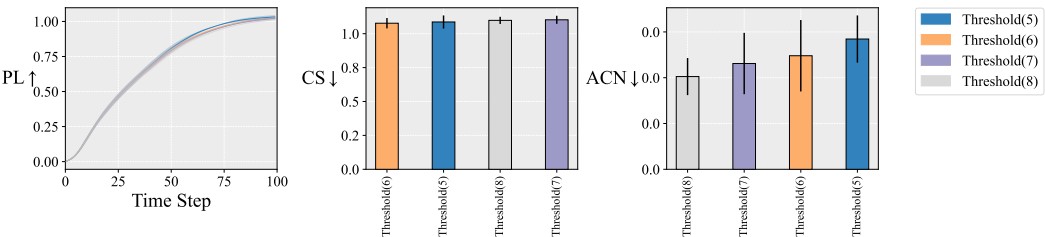

Figure 9: Ablation study results on the threshold $c'$. We fix the threshold $c' = 8$ and evaluate DualGF with different initial lambda.

### D.2  ABLATIONS ON THE SIZE OF THE EXAMPLE SETS

This ablation study is conducted on *Clustering*. For simplicity, we just fix the $\lambda_t$ at the initial value.

**Target Set:** We replace the target score network with ablated versions trained on different scales of the target set. As shown in the figure. 10, the performance of DualGF does not significantly drop with the number of target examples decreasing. Notably, with only 100 target examples, *DualGF(Tar 1e2)* achieves comparable performance to the *DualGF(Tar 1e5)* in PL, CS and ACN.

**Support Set:** Similarly, we replace the support score network with ablated versions trained on different scales of the support set. As shown in the figure. 11, the performance of DualGF significantly

drops after the number of support examples is smaller than 50,000. This is reasonable since the volume of the free space increases exponentially with the dimension of the state space. Thus, it is infeasible to learn a good support score network with few examples. In our cases, using 50,000 to 100,000 support examples works well for 42-dimensional state space (*i.e.*, 21 balls).

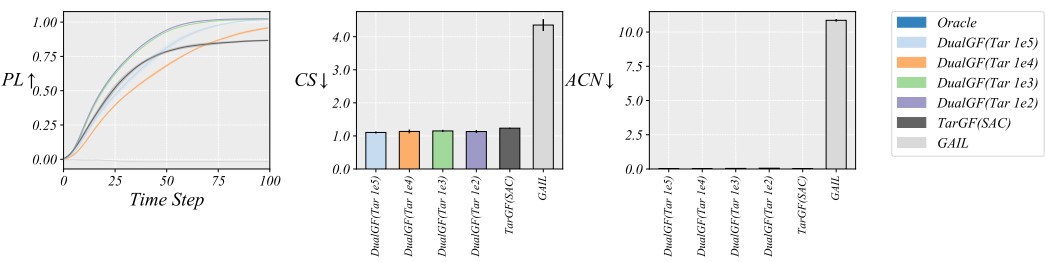

Figure 10: Ablation results on the number of target examples.

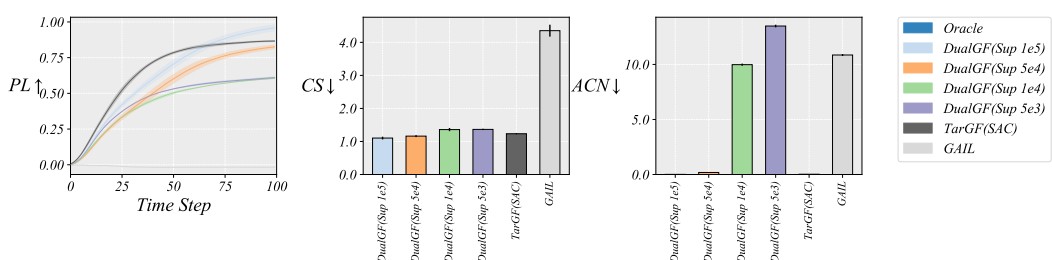

Figure 11: Ablation results on the number of support examples.

### D.3 ABLATIONS ON THE CHOICE OF NOISE LEVEL OF THE SCORE NETWORK

This ablation study is conducted on *Clustering*. For simplicity, we just fix the $\lambda_t$ at the initial value.

**Target Score Network:** We fix the noise scale of the support score network at $t = 0.01$ (same as DualGF in the main paper). We then compare the performance of the target score networks conditioned on different levels of noise scale in Figure. 12. With the noise level decreasing, the performance on PL and ACN slightly drops. This indicates the target score network can still provide useful target gradients under smaller noise levels. However, we observe that the target score network conditioned on a smaller noise scale seems to be more 'unsafe'. Thus, we have to specify a larger initial value of $\lambda_t$ to regularise the target gradient. For $t = 0.005, 0.001, 0.0005, 0.0001$, we specify $\lambda_t = 4, 10, 15, 20$ respectively.

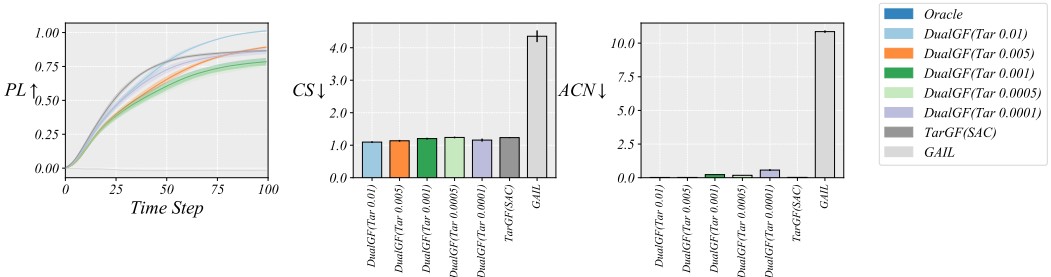

Figure 12: Ablation results on the noise level of the target score network.

**Support Score Network:** Similarly, we fix the noise scale of the target score network at $t = 0.01$ (same as DualGF in the main paper) and $\lambda_t = 3.0$ for all experiments. We then compare the performance of the support score networks conditioned on different levels of noise scale in Figure. 13. With the noise level decreasing, the performance on PL significantly drops while the ACN remains comparable to the original DualGF. In practice, we observe that the target score network conditioned on a smaller noise scale tends to be more 'conservative', *i.e.*, the support gradients are of large magnitude so that they overwhelm the target gradients.

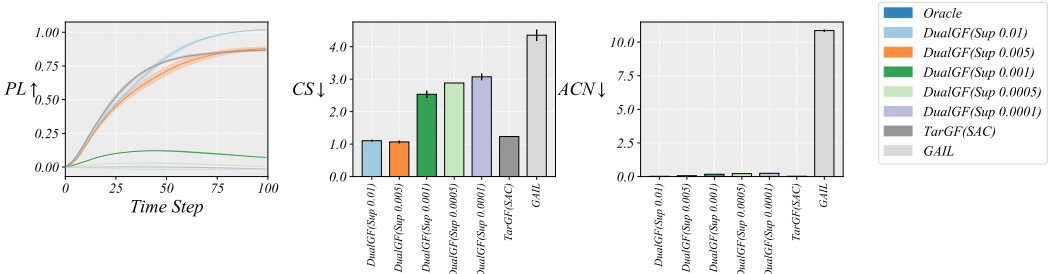

Figure 13: Ablation results on the noise level of the support score network.

## D.4 IMAGE-BASED DUALGF

The conditional state $\mathbf{s}_c$ can also be represented as an observation from the environment, *e.g.*, an RGB image. In Figure. 14, we demonstrate the image-based implementation of DualGF. Both the target and support network take an image $O_t \in \mathbb{R}^{64*64*3}$ and a agent-state $\mathbf{s}_a^t = [x_a, y_a] \in \mathbb{R}^2$ as input and output a gradient $\mathbf{\Phi}_{tar}^\theta(\mathbf{s}_a^t, O_t) \in \mathbb{R}^2$ located on the agent-state. The image is simply encoded by three layers of CNN and then reshaped to a feature vector. This vector is then concatenated with the agent's feature vector to be fed into a final output linear layer.

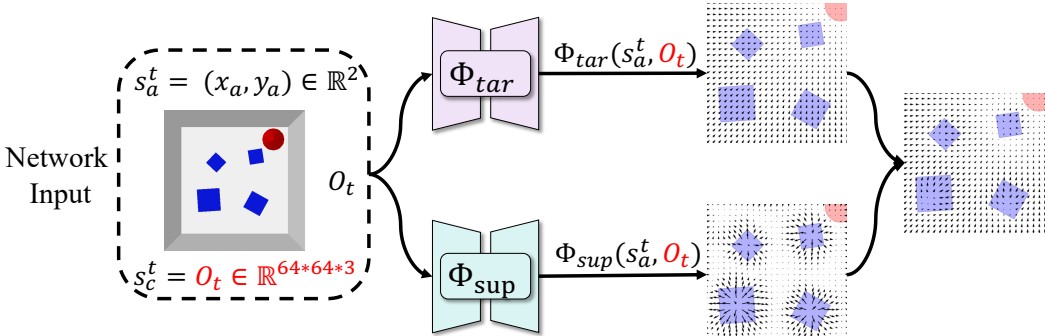

Figure 14: The image-based implementation of DualGF on *Navigation* and *Tracking*.

We compare this image-based version (*i.e.*, *DualGF(Image)*) with learning-based baselines. Notably, these baselines still take the low-dimensional state as input. As shown in Figure. 3, *DualGF(Image)* outperforms all the baselines across all the metrics. This indicates our method is still effective when the state representation is complicated, such as pixel input.

## D.5 SUCCESS RATE ON BALL REARRANGEMENT AND ROOM REARRANGEMENT

We also report planning success rate (SR) on ball and room rearrangement. Since the number of objects in these environments is quite large, so we relax the success threshold to 'less than $N$ collisions in an episode' where $N$ denotes the number of objects in each episode. As shown in Figure. 15, Du-

Table 3: Comparative results on *Navigation* and *Tracking*. We compare the image-based implementation of DualGF with state-based baselines.

| | Navigation (Static) | | Navigation (Dynamic) | | Tracking | |
|---|---|---|---|---|---|---|
| | TRS ↑ | SR ↑ | TRS ↑ | SR ↑ | TRS ↑ | SR ↑ |
| RL (SAC) | $9.1 \pm 0.6$ | $0.21 \pm 0.04$ | $8.3 \pm 0.4$ | $0.19 \pm 0.01$ | $0.0 \pm 0.0$ | $0.21 \pm 0.02$ |
| GAIL | $1.5 \pm 0.4$ | $0.35 \pm 0.01$ | $1.6 \pm 1.0$ | $0.20 \pm 0.02$ | $0.1 \pm 0.1$ | $0.18 \pm 0.04$ |
| RCE | $2.2 \pm 1.2$ | $0.13 \pm 0.01$ | $2.2 \pm 0.4$ | $0.22 \pm 0.04$ | $0.1 \pm 0.0$ | $0.17 \pm 0.03$ |
| TarGF (SAC) | $7.0 \pm 0.2$ | $0.19 \pm 0.02$ | $4.4 \pm 0.3$ | $0.19 \pm 0.01$ | $0.2 \pm 0.1$ | $0.18 \pm 0.04$ |
| DualGF (Image) | $\mathbf{16.4 \pm 1.0}$ | $\mathbf{0.24 \pm 0.02}$ | $\mathbf{15.7 \pm 2.7}$ | $\mathbf{0.26 \pm 0.02}$ | $\mathbf{0.4 \pm 0.0}$ | $\mathbf{0.27 \pm 0.01}$ |

alGF achieves comparable performance with model-based approach *TarGF(ORCA)* and outperforms all the other learning-based baselines except for *Room Rearrangement*.

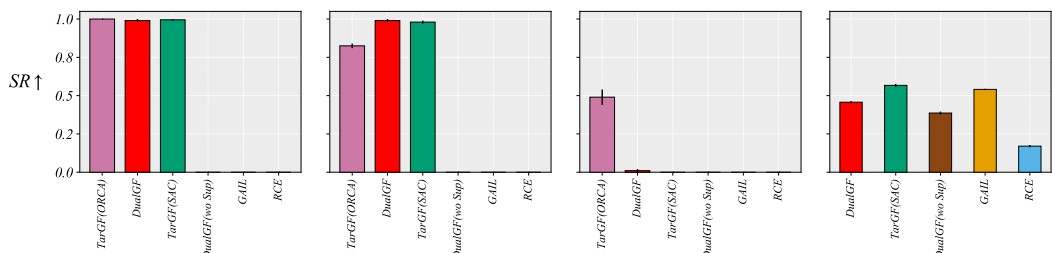

Figure 15: Success rates of all methods. From left to right, we present results on *Circling*, *Clustering*, *Circling+Clustering* and *Room* respectively.

## D.6 FAILURES: LOCAL MINIMUM PROBLEM

Similar to Lee & Park (2003), the most typical failure of DualGF is the local minimum problem. Figure. 16 demonstrate this problem in *Navigation* and *Room Rearrangement*.

On the left side, there are some regions in the dual gradient field where the target gradient and the support gradient cancel each other. Besides, the dual gradients around these regions point back to them. Once the agent steps into these regions, it will get stuck and cannot get out.

On the right side, we demonstrate another implicit type of this problem, where the agent is not stuck by the obstacle but by itself. For instance, the target gradient leads a table and a shelf to move in the opposite direction while the support gradient provides a 'repulsive force' on them. Thus, both the shelf and the table get stuck in the left corner.

One potential solution is to leverage test-time sampling to increase the optimality of the planning results.

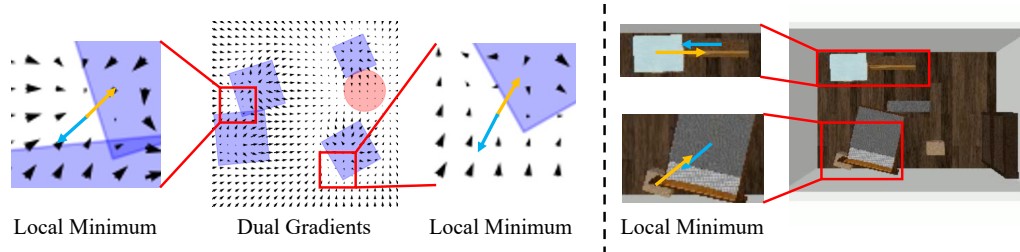

Local Minimum    Dual Gradients    Local Minimum    Local Minimum

Figure 16: The local minimum problem of DualGF. **Left:** Case study on *Ball Rearrangement*. **Right:** Case study on *Room Rearrangement*.

### D.7 MITIGATING THE INSTABILITY OF LAGRANGIAN UPDATES

The inherent instability of the Lagrangian updates may lead to bad results as shown in Figure. 17 (a) and (b): When the $\lambda_t$ increases too fast, the support gradients overwhelm the target gradients so the balls are too 'conservative' and do not move to the target distribution. When the $\lambda_t$ drops too fast, the target gradients overwhelm the support gradients so that the balls are too 'aggressive' and collide with each other.

We can mitigate this problem by setting a small learning rate and clipping the gradient. When $\lambda_t$ is properly updated, we can obtain appealing results as shown in Figure. 17 (c).

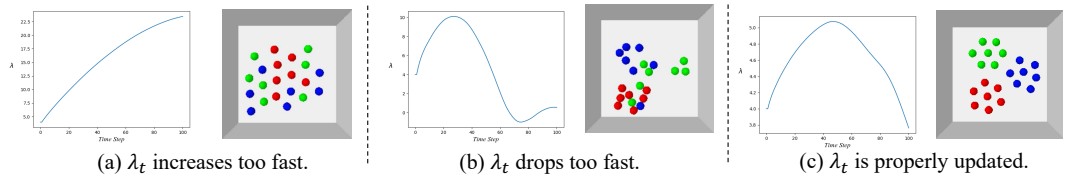

(a) $\lambda_t$ increases too fast.      (b) $\lambda_t$ drops too fast.      (c) $\lambda_t$ is properly updated.

Figure 17: Success and failure cases of Lagrangian updates.

### D.8 ANALYSE THE GRADIENT OF THE PERTURBED DISTRIBUTION

Our score networks trained via denoising score matching are essentially matching the score of the perturbed distribution $\nabla_{\mathbf{s}} \log p_\sigma(\mathbf{s})$ where

$$p_\sigma(\hat{\mathbf{s}}) = \int q_\sigma(\hat{\mathbf{s}}|\mathbf{s})p(\mathbf{s})d\mathbf{s}$$

$$\nabla_{\hat{\mathbf{s}}}p_\sigma(\hat{\mathbf{s}}) = \int \nabla_{\hat{\mathbf{s}}}q_\sigma(\hat{\mathbf{s}}|\mathbf{s})p(\mathbf{s})d\mathbf{s} \tag{17}$$

When $q_\sigma$ is a Gaussian kernel, we can derive the closed form of $\nabla_{\hat{\mathbf{s}}}q_\sigma(\hat{\mathbf{s}}|\mathbf{s})$:

$$\nabla_{\hat{\mathbf{s}}}q_\sigma(\hat{\mathbf{s}}|\mathbf{s}) = C(\mathbf{s}, \hat{\mathbf{s}}) \cdot (\mathbf{s} - \hat{\mathbf{s}}), C(\mathbf{s}, \hat{\mathbf{s}}) = \frac{e^{-\frac{||\mathbf{s}-\hat{\mathbf{s}}||_2}{2\sigma^2}}}{\sqrt{2\pi}\sigma} \tag{18}$$

where $C(\mathbf{s}, \hat{\mathbf{s}}) > 0$ is a scalar function depends on $\mathbf{s}$ and $\hat{\mathbf{s}}$. Thus, $\nabla_{\hat{\mathbf{s}}}q_\sigma(\hat{\mathbf{s}}|\mathbf{s})$ essentially points to the *denoising direction*, *i.e.*, $\mathbf{s} - \hat{\mathbf{s}}$. Further, we have:

$$\nabla_{\hat{\mathbf{s}}}p_\sigma(\hat{\mathbf{s}}) = \underbrace{\int C(\mathbf{s}, \hat{\mathbf{s}})p(\mathbf{s}) \cdot}_{averaged} \underbrace{(\mathbf{s} - \hat{\mathbf{s}})}_{denoising\ direction} ds \tag{19}$$

Intuitively, when $\hat{\mathbf{s}}$ lies in the zero-density region of $p(\mathbf{s})$, *i.e.*, $p(\hat{\mathbf{s}}) = 0$, the gradient of the perturbed distribution $\nabla_{\hat{\mathbf{s}}}p_\sigma(\hat{\mathbf{s}})$ still points to the *averaged denoising direction* to the positive-density region.

To summarise, the *score* of the perturbed distribution $\nabla_{\mathbf{s}} \log p_\sigma(\mathbf{s})$ is in the same direction with $\nabla_{\mathbf{s}}p_\sigma(\mathbf{s})$:

$$\nabla_{\mathbf{s}} \log p_\sigma(\mathbf{s}) = \frac{1}{p_\sigma(\mathbf{s})} \cdot \nabla_{\mathbf{s}}p_\sigma(\mathbf{s}) \tag{20}$$

where $\frac{1}{p_\sigma(\mathbf{s})} > 0$ obviously. Thus, in the zero-density region of the original distribution, the score networks are estimating *the averaged denoising direction* to the positive-density region.

### D.9 THE LIMITATION OF MANUALLY DESIGNING THE REWARD FOR BALL REARRANGEMENT

Here we take *Circling* for example to demonstrate the non-triviality of example-based planning. We leverage the pseudo-likelihood function of *Circling* as a manually designed reward function to train an RL agent, namely *Heuristic(SAC)*.

As shown in Figure. 18, our method achieves significantly higher PL than *Heuristic(SAC)*, which indicates our method is better in both sample quality and efficiency. Our method is also better than *Heuristic(SAC)* in CS, which indicates our method is better in diversity and mode coverage.

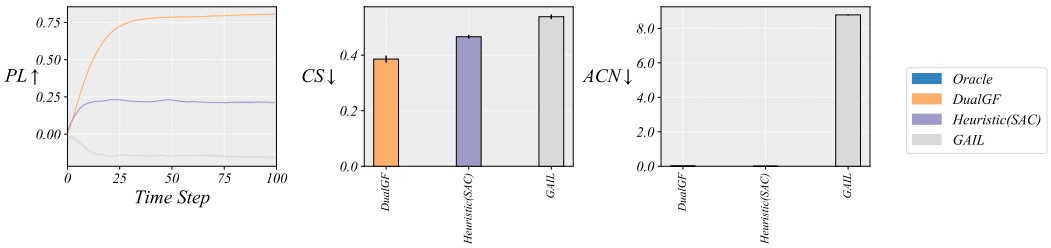

Figure 18: Comparison between *DualGF* and RL agent trained by manually designed reward. The experiments are conducted on *Circling*.

## D.10 EVALUATION ON ONLY SUCCESS TRAJECTORIES

In *Navigation* and *Tracking*, we report task return (TR) on non-collision trajectories. As shown in Table. 4, our method still outperforms the learning-based baselines across all the tasks and metrics.

However, this evaluation will also be problematic since the success trajectories of different methods are of different difficulties. For instance, the potential field baseline is of only 28% Success Rate yet achieves the highest Task Return. However, we observe that this method only succeeds when the initial state is quite near to the goal which is easier for the agent to obtain a higher task return.

Table 4: Comparative results on *Navigation* and *Tracking*. We report task return (TR) on non-collision trajectories.

|  |  | Navigation (Static) | | Navigation (Dynamic) | | Tracking | |
|---|---|---|---|---|---|---|---|
|  |  | TR ↑ | SR ↑ | TR ↑ | SR ↑ | TR ↑ | SR ↑ |
| Learning Based | RL (SAC) | $44.4 \pm 5.7$ | $0.21 \pm 0.04$ | $42.4 \pm 0.8$ | $0.19 \pm 0.01$ | $0.2 \pm 0.0$ | $0.21 \pm 0.02$ |
|  | GAIL | $4.4 \pm 1.1$ | $0.35 \pm 0.01$ | $7.5 \pm 4.3$ | $0.20 \pm 0.02$ | $0.6 \pm 0.5$ | $0.18 \pm 0.04$ |
|  | RCE | $15.7 \pm 7.9$ | $0.13 \pm 0.01$ | $9.8 \pm 0.1$ | $0.22 \pm 0.04$ | $0.4 \pm 0.1$ | $0.17 \pm 0.03$ |
|  | TarGF (SAC) | $36.2 \pm 3.9$ | $0.19 \pm 0.02$ | $0.7 \pm 0.4$ | $0.19 \pm 0.01$ | $0.1 \pm 0.0$ | $0.18 \pm 0.04$ |
|  | DualGF (wo Sup) | $0.0 \pm 0.0$ | $0.07 \pm 0.01$ | $13.1 \pm 4.9$ | $0.09 \pm 0.03$ | $0.2 \pm 0.06$ | $0.11 \pm 0.04$ |
|  | DualGF | $\mathbf{81.3 \pm 0.4}$ | $\mathbf{0.37 \pm 0.02}$ | $\mathbf{66.9 \pm 7.3}$ | $\mathbf{0.39 \pm 0.05}$ | $\mathbf{6.0 \pm 0.5}$ | $\mathbf{0.56 \pm 0.03}$ |
| Planning Based | PRM (Reference) | $85.0 \pm 0.4$ | $0.45 \pm 0.04$ | $84.0 \pm 0.1$ | $0.47 \pm 0.04$ | $25.7 \pm 0.4$ | $0.16 \pm 0.04$ |
|  | Potential Field | $77.4 \pm 1.8$ | $0.45 \pm 0.00$ | $84.3 \pm 0.3$ | $0.28 \pm 0.07$ | $22.4 \pm 2.4$ | $0.09 \pm 0.02$ |

