# OpenReview forum: "Example-based Planning via Dual Gradient Fields"
_ICLR.cc/2023/Conference — Submitted to ICLR 2023_

### Official Review · Reviewer_kjqZ · 2022-10-16

**Confidence:** 4
**Correctness:** 3
**Technical Novelty And Significance:** 2
**Empirical Novelty And Significance:** 2
**Recommendation:** 3

**Clarity, Quality, Novelty And Reproducibility:**

The paper was mostly clear up until the experimentation section. This section felt rushed with several unclear points (see details below). I think the idea of extracting gradient for both task and the environment limitations is novel and interesting.

Details:
- How did you "manually designing the reward function": Can you provide more discussion of why this is hard. For your domains, they did not seem to be difficult. I also could not find the exact reward values for all cases.
- "free space, i.e.states without collision" => Add space after dot
- "π : S → S", Shouldn't the policy output an action rather than the next state? I understand your are using a simple controller but the definition this way hides this assumption.
- "Online interactions are costly": But all of your environments are simulated so simulations are free.
- "Futher" => Further
- r_t reward and r_{i,j} for collision is confusing
- "Heuristic-based Baseline: Oracle: We perturb the target examples via a small Gaussian noise to obtain the terminal states at an oracle level." Not clear what you mean.
- "Task Reward" : You meant Task return as it is the cumulative values of rewards.
- Averaged Collision Number: They definition is confusing, but if I understood it correctly, it means the average number of objects colliding on each step of the plan. To me this is a hard constraint and if violated the rest of plan does not matter. You should filter the results that do not respect this and provide insight into the remaining valid paths based on other metrics. Also r_i,j should have t subscript as well as object are moving through time.
- In Figure 3, can you keep the X and Y axis values the same across figures (i.e. same algorithm order on X axis and same scale used for Y axis)? It will help readability immensely.
- Bring images later in the paper. Currently it is distracting that they are all presented before the main text corresponding to them
- Figure 5, PL row, Clustering: What does it mean to have likelihood numbers more than 1?
- Why using various metrics? PL specifically is not a great metric, as almost getting to the goal is not the same as getting to the goal. ACN should be zero for good paths. Once you found good paths then you can look at their average return and coverage score.
- Not sure why you needed ablation studies to remove support gradient as it captures your environment limitation.

**Strength And Weaknesses:**

Strengths
+ The idea of separating the task from the world limitations is great. The latter can be generalized across various tasks.
+ The whole idea is simple and easy to implement, although the assumption of having access to the P_tar may not be easy for all cases.
+ The approach is overall explained well.

Weaknesses
- Experimentations was the main low light of the paper. Both the methodology and the clarity needs more attention. Regarding the methodology, as part of solution, it is okay to bring the world constraints as soft penalties, yet it is not okay to change the problem based on your solution. If the world does not allow collisions, then merging the quality of colliding and non-colliding trajectories muddies the results, which in turn makes it difficult to judge the effectiveness of your method compared to others. For example, if your path collided in the beginning, it is not meaningful to discuss how close the end of that path was to the target distribution. A better metric to your ACN, is path feasibility: what percentage of generated paths were collision free. Concretely, I recommend focusing on TR (main) and CS (secondary) metrics for collision free paths. If an algorithm fails to find paths it should reflect it in the TR.

- Regarding limitations, while authors discussed the assumption of a simple controller that can move object(s) from s to s', they did not discuss about the situations were the combined gradient may not work well. For example, what if the two gradients cancel each other out and the agent freezes?

**Summary Of The Paper:**

Authors introduced Dual Gradient Field (Dual GF) as an offline learning example-based planning. The two gradients are coming from the target task and the support task. The former encourages the movement towards accomplishing the goal, while the latter enforces safety (e.g. avoiding collisions). Offline, two networks are trained to estimate each gradient using collected samples. During runtime, each gradients are calculated and mixed. The result is passed as an input to a heuristic controller that generates the movement. Authors evaluated their approach across three environments comparing with seven learning, planning, and oracle based techniques.

**Summary Of The Review:**

Overall, I think the paper is going in a great direction, but I believe authors should take a step back for their experimentations in order to turn this into an amazing publication.

---

### Official Review · Reviewer_QU9s · 2022-10-24

**Confidence:** 3
**Correctness:** 4
**Technical Novelty And Significance:** 3
**Empirical Novelty And Significance:** 3
**Recommendation:** 8

**Clarity, Quality, Novelty And Reproducibility:**

The paper is well written, clear and seems to be novel with a minor contribution. See above for more details on how to improve the quality. I have also noted below some typos and minor unclear points:

=== MINOR POINTS ===

* remain to require -> require
* design -> designed in page 1 and 2
* remove two () in page 2
* success -> successful in Related Work section (page 3)
* how do you determine \epsilon in eq. 1 ?
* why is log p_sup^{\sigma} - c is hard to estimate directly?
* any drawbacks of approximating log p_sup^{\sigma} with eq. (8). ?
* Section 5.2, = is extra it seems.
* "TarGF is model-based framework" -> what is the model like for the introduced tasks?
* Why does PRM need to search a path again when the agent reaches the goal?
* pg. 9 Ball Rearrangement: single mode? what are the modes like?
* pg. 9 Ours -> DualGF?
* Limitations and Future Work: extending the framework for 'bision-based control and the embodied robot" seems too general.

**Strength And Weaknesses:**

* The strength of the paper is in showing how the proposed simple solution works across multiple environments, when evaluated with multiple metrics and compared to multiple alternative approaches. Overall the experiments are extensive and ablation studies are included.

* The paper is well written and the approach is introduced clearly.

* As for weaknesses, I was missing some more details on the tasks and the environment details. Some example successful paths generated by the agent could have been included. Likewise, we would have profited from including some failure example paths, in order to appreciate better what the agent can achieve and when, in what circumstances.

* On a related note, we do not know when the introduced can fail from the paper. When it fails, does it fail slightly or terribly? The ablation studies show the effect of initial lambda as one potential source of issues, from which the lambda update seem to recover.

* We know that Lagrangian relaxation used here to solve the problems is not fool-proof at all. Some details on the potential issues would have been more illuminating.

* I did not read the appendix - if some of these 'improvements' of the main text are mentioned there then I'd suggest putting pointers to the relevant sections in the main text.

**Summary Of The Paper:**

This paper proposes a novel path planning approach that learns two gradient fields from two sets of task and support examples, as opposed to learning from whole trajectories or (inverse) reinforcement learning. These gradient fields are mixed at runtime using a heuristic velocity-based controller to generate feasible paths that lead the agent to the goal states in several simulated test scenarios. It is shown that the method outperforms other learning based path planners and is more efficient in finding solutions compared to sampling-based planners.

**Summary Of The Review:**

Overall I recommend the acceptance of this paper, it is well written and it introduces a path planning approach that seem to improve over several learning-based alternatives introduced recently in the literature. Note the weaknesses mentioned, although the work is evaluated extensively, we're missing the details of failure cases and why the method works well (across multiple environments).

---

### Official Review · Reviewer_vd31 · 2022-10-25

**Confidence:** 3
**Correctness:** 2
**Technical Novelty And Significance:** 3
**Empirical Novelty And Significance:** 3
**Recommendation:** 5

**Clarity, Quality, Novelty And Reproducibility:**

* The paper is easy to follow but there are some typos and blanks.

* It would be great if the order of baselines in Figure 5 is consistent across CS and ACN.

* The proposed approach introduces learning target and support gradient fields from examples and finding a path following the mixture of gradients, which is simple and novel.

* As the state and action spaces are complicated, it would be great to provide code for reproducing the results.

**Strength And Weaknesses:**

### Strengths

* The proposed method is simple but effective at finding a collision-free path.

* The experiments show better goal-reaching performance as well as fewer collisions in various 2D environments.

* The experiments show that the proposed method scales well with the number of manipulatable objects or agents.

### Weaknesses

* The core of the proposed method is learning a good target gradient field and a support gradient field. However, learning the dual gradient fields purely relies on the target examples and a dataset of randomly sampled states from the free space. The paper claims that the support examples are easy to collect by randomly initializing objects; however, collecting such data is challenging in more complex environments, such as the real world.

* The proposed method simply infers the target gradient field from a set of target states. The experiments are done with linear state spaces (e.g. x, y coordinates), where estimating the target gradient field from a few target examples is straightforward. However, when the state space is complex and nonlinear, such as pixels and non-smooth state changes, the learned target gradient field may not provide meaningful gradients for path planning.

* Moreover, the experiments are done in toyish environments with relatively large datasets for target states (100,000 states). Although the size of the support example data is not mentioned in the paper, to learn a reasonable support gradient field, it must need data that covers most of the states along potential solution paths. This can limit the scalability of the proposed method to general path planning problems.

**Summary Of The Paper:**

This paper proposes an offline learning approach for path planning. The proposed Dual Gradient Fields (DualGF) model the probability distributions of (1) target states (e.g. goal states) and (2) free space. Then, the problem of finding a path toward a target state while avoiding collision becomes simply following the gradients of the mixture of two probability distributions. The experiments show that the proposed method can find a path without access to online interactions and the ground truth model.


**Summary Of The Review:**

Overall, the proposed method is novel, simple, and effective. However, the experiments are done in path planning in 2D actions, which does not resolve the concerns about learning reasonable support gradient fields and path planning with non-linear state and action spaces. Adding experiments on a complex, non-linear environment would make this paper strong. But, as of now, I would recommend weak rejection.

---

### Official Review · Reviewer_Msq6 · 2022-10-28

**Confidence:** 4
**Correctness:** 3
**Technical Novelty And Significance:** 3
**Empirical Novelty And Significance:** 3
**Recommendation:** 6

**Clarity, Quality, Novelty And Reproducibility:**

Overall, the paper is well written and clear to understand. However, the paper lacks discussion of some important aspects, including its relation to mixture potential path planning, the effect of noise in the DSM calculation to make the approach work, and the conceptual relation between lambda and collision avoidance abilities of the approach (see more details in the Strength and Weaknesses section).
The approach has novelty as it allows learning from just state examples without the need for state samples (without the need for trajectory samples). However, it would be important to put this novelty in the context of the related traditional frameworks (see above).

In the experimental section, the authors do not describe the environmental settings (which is left to the appendix. In particular in the navigation example, however, it would be important to include some description of the environment complexity in the main paper to allow readers to better assess the significance of the performance results. Another item that is lacking in the experiment section is any description of the size and form of the example set for both the target and support fields. How those examples are derived and how many are sued would be very important information to assess to what degree the system was able to generalize. Similarly, knowing the structure of the score matching network used would be useful to assess generalizing capabilities vs potential overfitting. In a similar way, the similarity measure used for the epsilon ball neighborhood (in particular in terms of the conditional state features - obstacle states) would be useful as this would significantly influence the tradeoff between target and support gradients).
While the paper presents all the technical approach, lack of the parameters makes reproduction of the results difficult to achieve.

The paper also contains a few grammatical issues:
* First paragraph of the Introduction: "...are hard to design the objectives/reward with human priors." needs to be rewritten.
* Page 2, second paragraph: "..., which largely alleviate." is incomplete.
* In equation 5, there should be a separation between max and min.
* Page 7, CS section "... terminal states S_T=." seems to have some missing description.
* In the caption for Figure 6, "...Clusting..." should be "...Clustering..."


**Strength And Weaknesses:**

The paper presents a new approach to learn path planning from simple examples. A strength here is that as the examples are not trajectories but rather just target and freespace states, the freespace examples often transfer across tasks and can thus be shared for different task objectives in the same types of environments. While this does not apply to the target samples, which need to be specific to the task, they should still be easier to provide to the system than entire trajectories. This reduced effort in providing examples makes the approach interesting for a range of tasks where task objectives can easily be translated into target state examples.

Weaknesses of the paper lie mainly in the lack of discussion of some of the underlying assumptions in the approach and some missing discussions related to relations to other navigation planning work.
One implicit assumption that seems to be made (and is not discussed) is that the planning is applied to a holonomic system (i.e. a system with no inherent movement constraints. The reason for this is that in the problem statement section it seems too be assumed that any point in an epsilon ball around the current state is reachable by the system. This assumption makes section 4.3 (the integration with low-level control) problematic as it is not clear that a low-level control for a system with dynamic constraints exists for a step generated by the higher-level planner. It would be important to discuss this.
A second point that would warrant discussion is the seeming reliance on the right amount of noise in the DSM approach to make the target and support gradient fields work. In the technical description, the target gradient field is described as the gradient of the target state distribution. This, however seems to be very problematic since this gradient would be 0 anywhere outside the target region and thus not produce a gradient towards the target region. The only way to bridge this problem is the gaussian noise which effectively props up the value of the target potential (and thus the target gradient) for non-target points based on there proximity to the target region, basically creating a gradient towards the target region. It would be very useful for the authors to include some discussion regarding the way target gradients in non-target areas are derived (and how the same is achieved for non-freespace regions in the support gradient field). This effect also suggests that the choice of sigma in the DSM algorithm might have significant effect on the performance and a study of the effect of changing sigma would be very useful.

Overall, the approach seems to be similar in character to the mixture of goal and obstacle potential approaches to path planning just that the potentials are learned from samples and that the mixture is dynamically adjusted using the optimization of the Lagrangian formulation. Some discussion of this relation, and in particular a comparison with mixture potential path planning as an additional baseline would be very beneficial. In this context it would also be useful if the authors could compare navigation tasks in more complex environments (multi-room or maze type environments) to study to what degree their methodology to adjust mixture weights (lambda) dynamically overcomes the local minima problem of traditional mixture potential approaches, and to what degree it introduces collision problems due to target gradients overwhelming support gradients near obstacles due to lambda being too small. In this context it might also be beneficial to motivate why only a single step of lambda adjustment is made for each navigation planning step.


**Summary Of The Paper:**

This paper presents an approach to learn target (goal) and support (obstacle) gradient fields for navigation planning from a set of example configurations. As opposed to many other learning approaches for navigation planning, it does not rely on demonstration trajectories but only on sets of successful target configurations and of freespace configurations. Based on these gradient fields, trained in the form of score networks, a Lagrangian formulation is used to derive a planning algorithm that dynamically mixes the gradient fields to successfully navigate to a target in a new configuration. Studies show that the system can perform well in terms of task success with relatively low computational cost compared to traditional randomized path planners and can address complex re-arranging tasks.
The main contribution of the paper lies in presenting an approach that can build a path planner for a task from just a set of example configurations. The studies performed show that this simple approach can be effective at planning for a range of tasks with relatively limited overhead.

**Summary Of The Review:**

The paper presents an interesting approach with one novel that attempts to learn a gradient based planning system from state examples without the need for rewards or trajectories. The usability of the approach is demonstrated in a number of examples with tasks of different complexity.
The paper, however, lacks discussion of some implicit assumptions and of its relation with traditional mixture potential path planning approaches (this seems important as it seems to be largely an adaptive mixture potential approach with dynamic mixture weight adjustment).
While the experiments cover interesting domains, their detailed description is completely left to the appendix with not even a short description of the obstacle complexity I the paper, making it hard to assess the results. In addition, it seems that a study of the influence of the noise parameter sigma would be of high importance as it seems to drive the propagation of the target gradient field away form the target/goal region.

---

### Decision · Program_Chairs · 2023-01-20

**Decision:**

Reject

**Justification For Why Not Higher Score:**

The paper is not quite above the bar with respect to:
- novelty wrt closely related paper
- acknowledging weaknesses of the method
- providing strong baselines and consistent benchmarks

**Justification For Why Not Lower Score:**

N/A

**Metareview: Summary, Strengths And Weaknesses:**

This paper provides a novel mechanism for generating robot motion.  It trains two neural networks, each of which maps a world configuration (including robot and objects whose poses can vary across problem instances) into a gradient vector for the robot.  One network represents a potential field oriented toward configurations that satisfy a goal condition; the other represents a potential field that repels the robot from obstacles.  These two fields are combined, using an adjustable mixture parameter, to generate robot velocities.   There is no input geometric or kinematic model.  Instead, goal configurations are indicated through a set of samples, and free-space configurations are indicated through another set of samples.

The method is applied to a single-robot path-planning problem as well as to two "rearrangement" problems, which are actually instances of a problem class more typically known as multi-robot path planning (the objects to be "rearranged" can move in parallel, under their own steam).  It is much more effective than generic reinforcement-learning methods that don't "know" they are solving path-planning problems, but in basic path planning, a classical RRT-based method can solve more problem instances, although much more slowly.

The idea of learning gradient fields is relatively new, although this paper is using a framing that is almost exactly the same as that of [1].  The novelty in this paper is the addition of the obstacle repulsion field and the balancing between the attractive and repulsive fields.

The work is interesting and might have some value in the robotics context, but there are several substantial concerns that make the paper not ready for publication.

1.  It was only after the reviewers pointed it out that the authors acknowledged the problem of local optima in using potential fields for robot motion control.  Potential fields were hugely popular in the 1980s and 90s, but has demonstrable weaknesses in even moderately complex domains.  Reference [2] outline the difficulties and provide a method for avoiding local optima (which is actually equivalent to computing a value function!).   The method in this paper for adapting the mixing parameter is interesting.  It would be important to study it in isolation first:  if the correct potential fields are known, what class of navigation problems can your method solve?  Ideally some theoretical, or possibly empirical analysis (for example, with U-shaped obstacles that "catch" the robot, or the classic "bug-trap") would strengthen this aspect of the paper.

Ultimately, it is not necessary to be able to solve all possible navigation problems efficiently, but it is important to clearly understand the strengths and weaknesses of this method and have a strategy for detecting failure and falling back to a slower but complete method (e.g., some form of planning or value iteration using your learned fields.)

2.  It seems that it will be difficult to get sufficient coverage of the free cspace in the training data in high-dimensional C spaces.   How does the number of required training samples relate to the density of the obstacles or the narrowness of the passages or some other measure of path-finding difficulty, as well as the dimensionality of the CSpace?

3.  The evaluation criteria seemed to be very inconsistent.  The presented approach as well as the RL methods were scored with a penalty for crashing (but the agent was allowed to continue) whereas the planning-based methods operated under the hard constraint of finding a collision-free path.  It is rare in robot motion generation that collisions are allowed.   The authors point out that it is more difficult to compare results when there is a hard constraint, but it's nothing like impossible.  One generally reports both the percentage of problems solved correctly (that is, without constraint violation) and then, among those, the solution quality and/or computation time required.

Note also that the learning-based method requires substantial off-line computation, to gather the data and train the networks.  It is "paid for" in amortization over all subsequent problem instances that need to be solved, but should at least be mentioned.  So, although the planning-based approaches take much more time to execute online, we might be interested to know the trade-off with respect to off-line computation.

4.  Finally, the planning-based baselines are weak.  There are much more efficient RRT/PRM impementations (see the OMPL library).  And, there is a huge literature on multi-robot path planning which is a much better way to approach your second two sets of test problems.

[1] Wu, Zhong, Xia, Dong: Targf: Learning target gradient field for object rearrangement.  NeurIPS, 2022.
[2] Rimon, Koditschek: Exact robot navigation using artificial potential functions. IEEE Tran Robotics and Automation, 1992.
[3] Svestka, Overmars: Coordinated path planning for multiple robots. Rob and Auton Sys, 1998.

**Summary Of Ac-Reviewer Meeting:**

Three of the reviewers were able to participate in a meeting.  We weighed the strengths and weaknesses in detail and all of the discussion is summarized in the meta-review above.